# MEDSAT: A Public Health Dataset for England Featuring Medical Prescriptions and Satellite Imagery

**Sanja Šćepanović**[1,2,†,∗]   **Ivica Obadic**[2,3,∗]   **Sagar Joglekar**[1,6,‡]   **Laura Giustarini**[5]
**Cristiano Nattero**[5]   **Daniele Quercia**[1,4]   **Xiao Xiang Zhu**[2,3]
[1]Nokia Bell Labs    [2]Technical University of Munich    [3]Munich Center for Machine Learning
[4]Centre for Urban Science and Progress, King's College London [5]WASDI    [6]Intercom

## Abstract

As extreme weather events become more frequent, understanding their impact on human health becomes increasingly crucial. However, the utilization of Earth Observation to effectively analyze the environmental context in relation to health remains limited. This limitation is primarily due to the lack of fine-grained spatial and temporal data in public and population health studies, hindering a comprehensive understanding of health outcomes. Additionally, obtaining appropriate environmental indices across different geographical levels and timeframes poses a challenge. For the years 2019 (pre-COVID) and 2020 (COVID), we collected spatio-temporal indicators for all Lower Layer Super Output Areas in England. These indicators included: i) 111 sociodemographic features linked to health in existing literature, ii) 43 environmental point features (e.g., greenery and air pollution levels), iii) 4 seasonal composite satellite images each with 11 bands, and iv) prescription prevalence associated with five medical conditions (depression, anxiety, diabetes, hypertension, and asthma), opioids and total prescriptions. We combined these indicators into a single MEDSAT dataset, the availability of which presents an opportunity for the machine learning community to develop new techniques specific to public health. These techniques would address challenges such as handling large and complex data volumes, performing effective feature engineering on environmental and sociodemographic factors, capturing spatial and temporal dependencies in the models, addressing imbalanced data distributions, developing novel computer vision methods for health modeling based on satellite imagery, ensuring model explainability, and achieving generalization beyond the specific geographical region.

## 1   Introduction

The impact of environmental factors on human health has gained significant attention in recent years, particularly in the face of increasing environmental pressures caused by extreme weather events. Understanding these impacts is crucial for effective public and population health interventions. However, existing studies often face challenges in obtaining appropriate health and environmental data.

Fine-grained and comprehensive data on the prevalence of medical conditions is often scarce in public health studies. Traditional approaches rely on infrequent surveys or cohort studies, such as NHANES [23], HSE [46], BRFSS [52], The Swiss National Cohort [60], or the UK Biobank [61]. However,

---

∗Shared first authorship.
†The work was done during the author's AI4EO Beyond Fellowship at the Technical University of Munich.
‡The work was done prior to the author's tenure at Intercom.

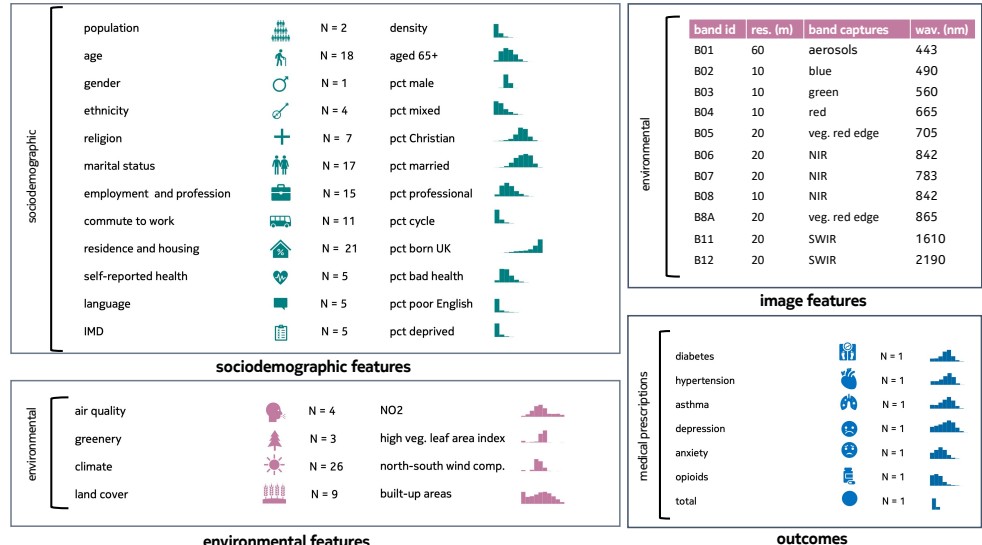

Figure 1: **Structure of MEDSAT dataset (single year):** This figure illustrates the four data components constituting MEDSAT: i) *sociodemographic* features (111), ii) *environmental* point features (43), iii) *image* features (4 seasonal Sentinel-2 composite tiles $\times$ 11 bands each), and iv) *prescription* outcomes (prevalence scores for 7 medical prescription types). The distributions of example variables are shown. Sociodemographic variables are presented as percentages ranging from 0 to 1, while environmental variables have varying ranges (e.g., NO2: (0-$3.1 \times 10^{-3}$) $mol/m^2$, $\mu = 2.43 \times 10^{-6}$, $\sigma = 2.0 \times 10^{-4}$). Outcome variables represent yearly prescription quantities per capita and are mostly normally distributed (except opioids and total prescriptions). For instance, diabetes prescriptions range from 0.02 to 104.84 ($\mu = 38.76$, $\sigma = 16.35$). Each Sentinel-2 composite image consists of 11 spectral bands. MEDSAT offers two such yearly snapshots, for 2019 and 2020. A comprehensive description can be found in the Appendix.

survey methods suffer from biases related to sampling, non-response, recall, and question wording. Cohort studies, while aiming to mitigate these biases, are limited in size, expensive, time-consuming, and prone to participant dropout. The All of Us Research Program [58] is an ambitious initiative recruiting over 1 million participants, but its representativeness and long-term engagement remain to be seen. In summary, existing health outcomes data are often limited in scope, granularity, and subject to various biases. Furthermore, despite the increasing availability of finer-resolution measurements for crucial environmental indicators relevant to public and population health, such as greenery, sun radiation, and air pollution, challenges persist in obtaining comprehensive and suitable indices that cover diverse geographical levels and timeframes. While the Earth Observation (EO) community has made significant efforts in capturing detailed satellite imagery with improved resolutions, frequencies, and accessibility, there remains a gap in transforming this vast amount of data into user-friendly indices that can be effectively utilized by non-technical stakeholders and the wider community unfamiliar with EO methods. Even when institutions provide data, such as air quality data from DEFRA [20] in the UK, individuals interested in compiling various environmental information often need to collect it from multiple sources, and there are spatial and temporal limitations to the available data.

In this paper, we present the MEDSAT dataset (Figure 1) consisting of four complementary components and covering two years (2019 and 2020), specifically designed for studying the effects of the environment on population health in small administrative areas in England. Our approach involves an open-source framework that utilizes National Health Services (NHS) practice-level prescription data to extract medical prescription prevalences at fine spatial (i.e., Lower Layer Super Output Area (LSOA)) and temporal (seasonal) scales for the entire population of 57 million. We derived environmental indicators from satellite products, such as Sentinel-5 and OMI, using Google Earth Engine [30]. Additionally, we created cloud-corrected seasonal composite images for 11 spectral bands of multispectral instrument (MSI) Sentinel-2 images in the WASDI platform [67], covering the whole of England, i.e., 130,279 km$^2$. Our dataset also includes 111 socioeconomic indicators

Table 1: A comparison of MEDSAT with similar datasets, some of which are not publicly available. NHS corresponds to the UK health care system and DHS to the Dutch healthcare system.

| dataset | health indicator(s) | indicator source | imagery | env. | soc. | spatial unit | public |
|---|---|---|---|---|---|---|---|
| SustainBench [69] | BMI, child mortality, water quality, sanitation | surveys | Landsat street view | ✗ | ✗ | village (59km2) | ✓ |
| Landscape Aesthetics [39] | environment scenicness | crowdsourcing | Sentinel-2 | ✓ | ✗ | 1.6km2 | ✓ |
| COVID-19 [62] | COVID-19 cases and deaths | WHO | ✗ | ✓ | ✓ | city | ✗ |
| Greenery & mortality [7] | mortality | Eurostat | ✗ | ✓ | ✓ | city | ✗ |
| Greenery & prescribing [31] | antidepressants **prescriptions** | DHS | ✗ | ✓ | ✓ | municipality (up to 506km2) | ✗ |
| Nat. env. & prescribing [27] | mortality, **prescriptions**: cardiovascular antidepressants | NHS | ✗ | ✓ | ✓ | LSOA (up to 18km2) | ✗ |
| **MEDSAT** | **prescriptions**: respiratory (asthma) metabolic (diabetes, hypertension) mental (depression, anxiety), opioids, & total | NHS | Sentinel-2 | ✓ | ✓ | LSOA (up to 18 km2) | ✓ |

obtained mostly from the UK census. By integrating these diverse datasets for the years 2019 and 2020, we provide researchers with a comprehensive resource for studying spatial and temporal health attributes and identifying regional health disparities.

## 2 Related Work

Environmental conditions such as air or noise pollution are relevant indicators for various health issues like asthma or heart diseases [22]. However, the benefits of using EO data to monitor the impact of environmental conditions on human health are still limited. This limitation stems from the existing datasets in the literature listed in Table 1 that either focus on a narrow set of conditions derived from surveys that might not be representative of the entire population or do not provide detailed medical prevalence data on a fine-grained spatial level. Often these datasets are not publicly available and fail to include environmental and sociodemographic features relevant to health studies. For example, the SustainBench dataset [69] presents the problem of predicting 4 different health indicators derived from surveys based on Landsat satellite imagery and street-view images. The spatial unit of this study corresponds to a village or a local community covering an area of $\approx 58$ km$^2$ and this dataset does not contain additional environmental and sociodemographic features. By considering a more abstract health indicator, Levering et al. [39] introduced a dataset that associates Sentinel-2 images with crowdsourced data for landscape scenicness used as a proxy for human health and well-being. By analyzing fine-grained geographical regions of 1.6 km$^2$, the authors discovered plausible associations between a landscape's beauty and its land cover distribution. Targeting a specific condition across the population, Temenos et al. [62] assembles a dataset that relates COVID-19 cases with point features describing environmental data for urban greenness, air quality, sociodemographic features, and health factors. The authors reveal that temperature and mobility trends are among the most important features for predicting COVID-19 cases. Yet, this dataset is not publicly available, does not contain any imagery and the point features represent entire cities, thus describing very coarse spatial units. Further, the impact of the natural environment on mortality rates and prescriptions for various conditions is investigated in [7, 31, 27]. However, the datasets used in these studies are also not publicly available, include only environmental variables related to greenery, and do not contain any imagery. Moreover, the analyses in [7, 31] are performed on larger spatial units like municipalities and cities.

In comparison to these works, our dataset, MEDSAT, enables comprehensive modeling of the population health on a very-fine-grained spatial level as it jointly offers environmental and sociodemographic features as well EO imagery at LSOA level. Further, it covers prescriptions associated with 5 medical conditions, as well as opioids and total, which allow a detailed understanding of specific condition-related factors, and shed light on the overall population health and well-being.

## 3 The MEDSAT Dataset

The MEDSAT dataset serves as a comprehensive resource for public and population health studies, encompassing medical prescription quantity per capita as outcomes and a wide array of sociodemographic, environmental and image features across 33K LSOAs in England (Figure 1). In this release,

we provide data snapshots for the years 2019 (pre-COVID) and 2020 (COVID). Sociodemographic variables align with the latest UK census from 2021. Figure 2 visualizes examples of variables present in MEDSAT.

Access the code at `https://github.com/sanja7s/MedSat`,
and the dataset at `https://doi.org/10.14459/2023mp1714817`.
The dataset is released under the CC BY-SA 4.0 license.

### 3.1 Sociodemographic Features

Our dataset comprises sociodemographic variables sourced from the latest UK Census in 2021 [1]. These variables encompass essential indicators employed in public and population health research, such as gender (percentage of males), age distribution (percentage within 5-year age groups up to 85 and above), deprivation scores (percentage of deprived households in 1-4 dimensions), self-reported health (percentage reporting health on a five-point scale), ethnicity (percentage of individuals with White, Asian, Black, or Mixed backgrounds), and English proficiency (percentage reporting English as their main language). Additional variables indirectly related to health outcomes were incorporated, covering religion, commute means and distance to work, residence and housing, profession, and marital status. Our sociodemographic data do not contain any personally identifiable information because census implements stringent privacy protection measures, including targeted record swapping and cell key perturbation, to ensure confidentiality without compromising aggregated statistics [1].

### 3.2 Environmental Features

We obtained environmental point features for the MEDSAT dataset using various satellite data products on Google Earth Engine (GEE) [30]. For *air quality*, we used satellite data products such as Sentinel-5P NRTI to derive nitrogen dioxide (NO2) [32], TOMS&OMI for ozone [4], and CAMS for total aerosols and PM2.5 [25]. *Greenery* variables were derived from Sentinel-2 MSI for NDVI and ERA5-ECMWF product for high/low vegetation greenery indices. *Climate* variables, including wind components, air temperature, soil temperature, atmospheric pressure, and incoming solar radiation, were obtained from ERA5-ECMWF. All *land cover* variables were sourced from Google Dynamics World product [11]. Importantly, our open-source code can be easily adapted to extract other similar indices, and at different *spatial* scales (e.g., wards or other countries) and *temporal* scales (e.g., monthly or different years). More information can be found in Appendix section D.2.

### 3.3 Image Features

In addition to the above-described approach for deriving preprocessed environmental point features, in our dataset we also included the spectral bands provided by the Sentinel-2 mission [3], thus resulting in a more comprehensive set of environmental image features. Concretely, we processed Sentinel-2 images for the years 2019 and 2020 through the WASDI [67] platform by calculating average values for each of the four meteorological seasons [49]. We focused on 11 specific bands (Figure 1) capturing relevant environmental factors, excluding the band B10 that is primarily used for cloud detection and water vapor mapping. To ensure data consistency and quality, we resampled the bands to a uniform resolution of 10 m, applied cloud masks to exclude affected pixels, and computed pixel-per-pixel averages over time under cloud-free conditions. The resulting composite images represent the typical environmental characteristics for each season (over 500 GB per year in total).

To explore the potential of the high-resolution Sentinel-2 imagery for modeling population health on a fine-grained spatiotemporal level, we extracted image features per LSOA for each meteorological season with the procedure depicted in Figure 4 in Appendix. Concretely, we used the shapefiles that describe LSOA's geographical location to crop the LSOA pixels from a seasonal Sentinel-2 image. Next, we extracted basic image features from the cropped LSOA pixels by performing the following 5 aggregations per Sentinel-2 band: *mean, stdev, min, max*, and *median*, thus resulting in 55 image features per LSOA per season.

### 3.4 Prescription Outcomes

**Extracting Prescriptions on an LSOA Level from NHS data.** The monthly practice-level prescribing data in England, provided by the National Health Services (NHS) since July 2010 [47],

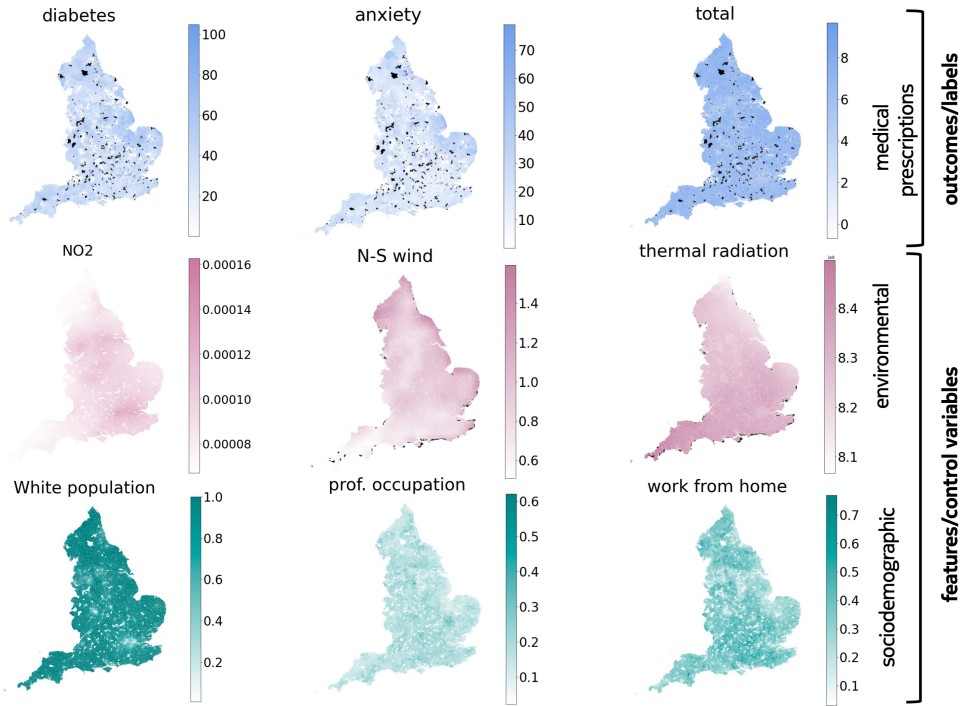

Figure 2: **Visualization of the MEDSAT point features.** The example distributions for the year 2020 of three health outcomes (diabetes, anxiety, and total prescriptions), three environmental variables (NO2, north-south wind component, and incoming thermal radiation), and three sociodemographic variables (percentage of White population, professional occupation, and work-from-home). The missing values are highlighted in black. Depending on the specific analyses intended, the missing value rate will be constrained by the outcomes, standing at 5.7%, and ranging up to 15.2% if all the features are to be used. Notably, we possess sociodemographic and image features data for all LSOAs. Please refer to Appendix for the details.

constitutes the foundation of our analysis. It offers anonymized information about monthly prescriptions across General Practitioner (GP) practices and patient membership to GP practices on an LSOA level. We parse this data to extract the prescribed drugs and summarize the total number of patients per LSOA and compute the fraction of a GP practice's patients associated with a specific LSOA. Further details about the NHS prescriptions data and the applied procedure for calculating the number of patients per LSOA can be found in the Appendix, Section D.4.

**Associating Prescriptions with a Condition.** To determine prescriptions related to specific medical conditions, our framework utilizes curated lists of drugs, such as the one collated for opioids by previous works [59, 18], or it leverages DrugBank [37] to automatically identify drugs associated with a given condition (see the Appendix DrugBank section D.4.3). DrugBank is an online database that provides comprehensive information on active pharmacological ingredients (APIs) and their corresponding conditions. Each drug name is associated with one or more conditions (i.e., symptoms and diseases), drug categories, and an Anatomical Therapeutic Chemical code assigned by the World Health Organization (WHO) for unique identification purposes. For instance, the drug name Citalopram (`https://go.drugbank.com/drugs/DB00215`) is linked to a range of diseases, including Depression, Anorexia Nervosa, Generalized Anxiety Disorder, and Post Traumatic Stress Disorder. During our crawl, we obtained data on 9,105 drug names from the website, and by filtering out drug names that were not linked to any drug categories, symptoms, or conditions we were left with 3,013 drug names. Next, we generated a curated list of drugs associated with a specific condition by selecting drugs from DrugBank that were linked to that condition (e.g., we associated Citalopram with depression, anxiety, and the other conditions mentioned above).

**Estimating the Prescription Prevalence.** To estimate the number of prescriptions associated with a specific condition $c$, we first matched the condition-specific drugs from DrugBank with their British National Formulary (BNF) codes from the NHS prescribing dataset. Next, the number of prescriptions for a specific condition $c$ in area $a$ is computed using the following formula:

$$N_c(a) = \sum_{GP \in a} N_c(GP) \cdot f(GP, a), \tag{1}$$

Here, $N_c(GP)$ represents the total number of prescriptions per GP for drugs associated with the curated list for the condition, and $f(GP, a)$ denotes the fraction of patients of the GP who reside in the area $a$. To ensure comparability across areas with varying population densities, we computed the metric of "prescriptions quantity per capita," commonly used in medical studies [19, 18], as follows:

$$\tilde{N}_c(a) = \frac{N_c(a)}{n_{pat}(a)}, \tag{2}$$

where $n_{pat}(a)$ corresponds to the total number of patients residing in the area $a$.

For MEDSAT we calculated medical prescriptions associated with three classes of conditions: i) *metabolic* (diabetes and hypertension), ii) *mental* (depression, and anxiety), and iii) *respiratory* (asthma); as well as *opioids* prescriptions, (which are predominantly prescribed for pain management, but they do have other applications, and have been associated with a consumption crisis in the UK [53, 56]), and *total prescriptions*, as a proxy for general health and well-being. We highlight that, for simplicity, we use condition names to refer to related prescriptions, however we cannot know for each individual prescription what was the exact cause for which it was prescribed. E.g., "depression prescriptions" means *antidepressants* and "anxiety prescriptions" means *anxiolytics*, regardless of actual use. Co-prescriptions across conditions may arise from this method, as it does not ascertain specific prescription reasons, as such details are absent in the NHS dataset. However, the multitude of studies examining prescriptions [59, 18, 8, 41, 34, 33, 31, 44, 35, 63, 12, 66, 29], akin to our approach, attests to its significance as a public health outcome.

Our prescription dataset does not contain any personally identifiable information, as it is derived from publicly available monthly prescription data provided at the level of practices, each serving numerous patients.

## 4 Results

### 4.1 Revealing Health Inequalities

In Figure 8 in Appendix, we present the healthcare accessibility disparities across regions. First, interestingly, we find a prevailing pattern where the number of registered patients exceeds the census population in most areas of the country. This aligns with previous investigations by UK authorities [64]. Second, although the correlation ($r = .87$, $p \approx 0$) is strong, certain LSOAs exhibit disproportionate patient-to-population ratios.

Additionally, our analysis highlights broader factors contributing to healthcare inequalities. The residual values, representing deviations from the linear fit of the patient to the population numbers, correlate with deprivation levels ($r = .16$, $p \approx 0$ for mid-deprived areas; $r = .22$, $p \approx 0$ for highly-deprived areas), suggesting a greater burden on healthcare access in socioeconomically disadvantaged regions. Moreover, the residual values exhibit a negative correlation ($r = -.40$, $p \approx 0$) with the percentage of White population, indicating disparities associated with ethnic backgrounds. For more details on this analysis, please refer to Section E.1 in Appendix.

### 4.2 Predicting Prescriptions

To evaluate the plausibility of our dataset for modeling population health, we applied a classical geostatistical method called Spatial Lag Model (SLM) [5] as well as trained the LightGBM machine learning model [36] to predict the medical prescriptions based on the point features, including image-derived ones, in our dataset. Both models were applied separately for every condition and a combination of the environmental, sociodemographic and image features, to better understand the contribution of different input features in modeling population health.

Table 6 in Appendix displays the SLM results. Collectively, the input features account for a variance ranging from 38% (for diabetes and total prescriptions) up to 63% (for opioids). Individually, sociodemographic features lead the way, explaining between 31% (total) and 52% (opioids) of the variance. They are followed by environmental point features, which account for variances from 13% (diabetes) to 49% (opioids). Image features, though least impactful, still cover a variance from 4% (diabetes) to 28% (opioids).

While the models such as SLM account for the well-known spatial autocorrelation effects [5] present in spatial analysis and modelling (referring to the process that creates clusters of values), machine learning models, such as LightGBM, require a special type of cross-validation that is adapted to account for these effects [55]. A block-buffered cross-validation is a common approach [38], and it is implemented in an R package called `blockCV` [65]. We employed this package to calculate spatial folds on the input of our LSOA shapefiles. For the details, please refer to Appendix E.2. The results obtained through spatial cross-validation using LightGBM are detailed in Table 2. In the initial row, it is evident that even fundamental image features exhibit predictive capability, albeit to a limited extent. Conversely, the subsequent two rows highlight that environmental and sociodemographic features offer improved explanatory power for the observed variances, outperforming image-based features. Furthermore, these feature categories display varying degrees of importance across different conditions. Environmental attributes notably enhance the predictive accuracy of depression, opioid prescriptions, asthma, and total prescriptions. Conversely, sociodemographic features prove more effective in accurately forecasting prescriptions for other medical conditions. Notably, the integration of both environmental and sociodemographic characteristics becomes pivotal for a holistic model of population health. This is exemplified by the last row, indicating that using both, the environmental and the sociodemographic features results in the best $R^2$ scores for all conditions under consideration. Moreover, for both the SLM and the LightGBM model, we observe a consistent pattern that the prescriptions for the mental conditions are predicted with higher accuracy than the ones for the other conditions. In Appendix Section E.2, we present a detailed overview of this experiment's setup and provide a comparison of the LightGBM model with a Feed-Forward Neural Network (FNN), which shows that the LightGBM model consistently outperforms the FNN.

Table 2: **The average $R^2$ scores resulting from the 5-fold spatial cross-validation of LightGBM.** These scores are computed across various prescription types and combinations of dataset features specifically for the year 2020.

| | metabolic | | mental | | respiratory | | |
| input | diabetes | hypertension | depression | anxiety | asthma | opioids | total |
|---|---|---|---|---|---|---|---|
| **Image** | 0.02 ±0.07 | 0.15 ±0.12 | 0.15 ±0.15 | 0.14 ±0.15 | 0.14 ±0.13 | 0.19 ±0.14 | 0.07 ±0.1 |
| **Env.** | 0.19 ±0.08 | 0.33 ±0.13 | 0.42 ±0.15 | 0.41 ±0.13 | 0.37 ±0.11 | 0.52 ±0.12 | 0.26 ±0.1 |
| **Soc.** | 0.26 ±0.1 | 0.37 ±0.11 | 0.41 ±0.15 | 0.39 ±0.12 | 0.32 ±0.13 | 0.47 ±0.14 | 0.22 ±0.11 |
| **Env. + Soc.** | 0.35 ±0.08 | 0.44 ±0.1 | 0.50 ±0.13 | 0.48 ±0.12 | 0.43 ±0.1 | 0.6 ±0.1 | 0.31 ±0.1 |

**Uncovering Health Factors** Understanding the impact of the environment and sociodemographic conditions on human health has been a focus of many studies [14, 2]. Our dataset offers new perspectives on such studies, as it allows the investigation of these relationships on a fine-grained spatial level, and for many conditions simultaneously. To shed light on these perspectives, we apply the SHAP [42] approach on the LightGBM models trained for prescription prediction. In Figure 3 we show the 10 most important features estimated by the SHAP algorithm for the models used to predict diabetes and total prescriptions. The left plot shows that the sociodemographic indicators describing the occupation, commute habits, migration, and ethnicity are highly relevant for predicting diabetes prescriptions. Specifically, the model establishes a linkage between lower prescription rates and LSOAs characterized by a substantial prevalence of professional occupations, a high number of people working from home, an active bicycle commuting trend, and a prominent student population. Conversely, increased prescription levels are associated with LSOAs having a high proportion of individuals of Asian ethnicity. When it comes to the environmental features, the east-west wind component and PM2.5 appear to be relevant indicators as higher diabetes prescriptions align with LSOAs characterized by an eastward wind pattern and heightened air pollution levels, as indicated by the PM2.5 metric. Additionally, bare soil evaporation and ozone stand out as noteworthy contributors to diabetes prescriptions even though these features exhibit no straightforward linear association with their corresponding SHAP values. On the other hand, in the right plot, we note that the environmental features describing canopy evaporation, NO2, east-west wind, and thermal radiation rank among the

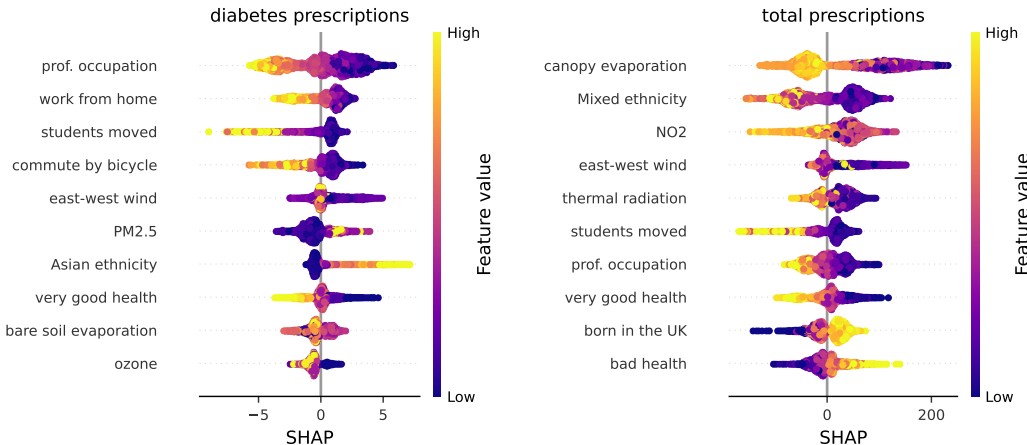

Figure 3: **SHAP summary plots for diabetes (left) and total prescriptions (right)**. The SHAP value for a feature indicates its contribution towards the difference between the prediction for an instance and the average model prediction. These plots reveal the 10 most important features for both conditions and the association between the feature values and their importance. In line with the results shown in Table 2, the sociodemographic indicators appear to be more relevant for modeling diabetes prescriptions than the environmental features while the opposite is observed for the total prescriptions. Although a similar set of features are ranked among the most important for both conditions, some features are particularly relevant for specific conditions, such as work from home and Asian ethnicity for diabetes prescriptions and thermal radiation and Mixed ethnicity for the total prescriptions.

top-5 most relevant features for estimating total prescriptions. Concretely, high values for canopy evaporation, NO2, and thermal radiation are negatively correlated with the total prescriptions while east-west wind displays a similar association as for the diabetes prescriptions. With respect to sociodemographic features, LSOAs characterized by mixed ethnicity are associated with a lower number of total prescriptions compared to the ones where the majority of the people are born in the UK. Moreover, we also see that low prescriptions are again associated with a high percentage of students and professional occupations. Finally, for both conditions, we notice that the positive self-assessment of health is linked to lower prescription values. Examples of SHAP values for the other conditions as well as dependence plots describing the feature interactions are provided in Sections E.2 in the Appendix.

**Describing Health of Environment Using Visual Concepts.**    To shed light on the potential benefits of using the Sentinel-2 imagery for modeling population health, we reveal the learned visual features patterns in Figure 4 by visualizing examples of LSOA Sentinel-2 images for which the LightGBM model trained on the simple image features closely approximates the actual opioids prescriptions. The LSOAs were visualized with the band combination (B11, B06 and B01) as these bands appeared among the most salient image bands according to the SHAP values for the LightGBM model shown in Appendix, Figure 13. First, we note that although the simple image features do not encode the size of an LSOA, they still enable the LightGBM model to associate higher opioid prescriptions with LSOAs covering larger geographical areas. Larger-area LSOAs are rural (because these administrative units are designed to have roughly equal populations), and it is known from previous research that opioid consumption is higher in rural areas [16]. Equally important, we also note that the LSOAs with high prescription values in the first row are characterized by a stronger presence of blue and pink colors occurring near traffic roads than those LSOAs with low opioid prescriptions in the bottom row. Due to the chosen band combination that displays the aerosols (B01) band in blue, this finding points out that the model can relate increased opioid prescriptions for LSOAs exposed to air pollutants. In conclusion, as observed in Table 2 and Table 6 in the Appendix, the environmental and sociodemographic features explain a higher percentage of the variance for prescription predictions. However, this analysis underscores that even basic image features can offer valuable insights into the intricate task of population health modeling. This suggests that leveraging the deep learning methodologies for processing the Sentinel-2 images at the LSOA level has great potential to improve

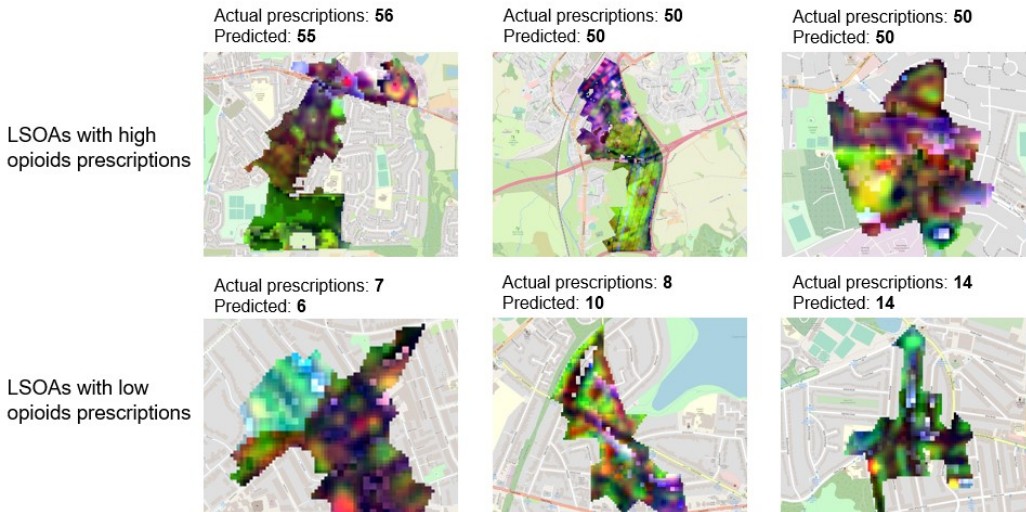

Actual prescriptions: **56**
Predicted: **55**

Actual prescriptions: **50**
Predicted: **50**

Actual prescriptions: **50**
Predicted: **50**

LSOAs with high opioids prescriptions

Actual prescriptions: **7**
Predicted: **6**

Actual prescriptions: **8**
Predicted: **10**

Actual prescriptions: **14**
Predicted: **14**

LSOAs with low opioids prescriptions

Figure 4: **Visualizing LSOA Instances in the (B11, B06, B01) band combination**. The short-wave infrared band (B11) is shown in red, the near-infrared band (B06) is shown in green and the aerosols band (B01) is shown in blue. The LSOAs with high and low opioid prescriptions are shown in the first and the second row, respectively. Remarkably, LSOAs with high opioid prescriptions cover a larger geographical area than those with low prescriptions (notice the higher zoom-in level), and have greater presence of aerosols (band B01) depicted in blue and purple colors.

prescription predictions by capturing the spatial dependencies inside an LSOA. Moreover, using the recent works in eXplainable Artificial Intelligence (xAI) such as [26] can portray environmental health through rich visual concepts, thus opening possibilities for novel insights about the relevant urban and rural structures influencing population health.

### 4.3 Temporal Analyses

Using MEDSAT, we analyzed temporal differences in outcome and environmental features between 2019 and 2020. Appendix Figures 15 and 16 illustrate varied distributions for both prescription quantities and environmental features. Notably, in the first COVID year (2020), there was a rise in prescriptions for *anxiety* and *depression* (in line with reports that the pandemic presented enormous challenges to mental health services in UK [13]) and *diabetes* medications in England, while *asthma* and *hypertension* prescriptions decreased [13].

The environmental shifts during the initial COVID year are evident from altered air pollutant distributions. Satellite data showed decreased levels of *NO2*, *ozone*, and *PM2.5* in England for 2020 (as reported erlier in [57]). Land cover changes saw increased *built* areas, likely due to heightened construction activity in the latter half of the year [24, 21], and reduced *trees* cover. Furthermore, 2020 witnessed elevated *temperatures*, *solar radiation*, and both components of *wind* compared to 2019.

## 5 Impact, Limitations, and Perspectives

We introduced MEDSAT, a unique dataset providing a comprehensive view of medical prescriptions, average yearly environmental indicators, image features, and sociodemographic factors across England for 2019 (pre-COVID) and 2020 (COVID). This resource enables a thorough assessment of health status for various conditions and exploration of their relationships with sociodemographic and environmental factors.

MEDSAT has three significant impacts. First, it has the potential to empower the development of novel machine learning (ML) approaches tailored for spatially-autocorrelated public health data [45], that can augment still predominant traditional statistical models like spatial linear regression [6] and BYM [9], as recent work indicates for xAI models [40]. MEDSAT enables ML research with large and complex data, effective feature engineering, capturing temporal dependencies, addressing

imbalanced data, ensuring interpretability, and achieving generalization across diverse regions. Secondly, MEDSAT can facilitate novel discoveries in public health by revealing influential factors that profoundly affect health outcomes. Through SHAP analyses, we confirmed the established link between diabetes and ethnicity, with higher prevalence among people of Asian descent [15, 28], and the preventive effects of biking and active commuting against diabetes and metabolic conditions [54]. Notably, our data from the initial year of the COVID-19 pandemic highlights the impact of socioeconomic factors. Higher percentages of professional occupations and individuals working from home are associated with lower prevalence of diabetes and total prescriptions, underscoring the influence of deprivation on health outcomes [10, 51, 68, 43]. Our findings not only confirm existing knowledge but also expose less-explored connections between the environment and human health. For instance, our SHAP results demonstrated associations between ozone exposure and mental health prescriptions, as well as between total aerosols and metabolic condition prescriptions. Furthermore, a north-sound wind is linked to a decrease in both types of prescriptions. Thirdly, MEDSAT enables groundbreaking discoveries in population health, particularly regarding health inequalities. Our preliminary analysis uncovers disparities in health accessibility among different economic and ethnic groups. By examining deprivation dimensions such as income, education, and occupational factors, across prescriptions of different types, we can gain a deeper understanding of their contributions to health disparities.

Although MEDSAT is among the most comprehensive publicly-available public and population health datasets to date, it is not without limitations. First, prescription prevalence may not always reflect the true prevalence of the medical condition itself. That is because disparities in healthcare access, privilege, knowledge, and stigmatization can influence prescription rates for certain conditions among different populations [50, 48, 17]. However, it is crucial to note that despite this limitation, our dataset offers a unique opportunity to disentangle these effects, especially when combined with other types of health outcome indicators. Compared to surveys and population samples, which come with their own set of biases, MEDSAT provides a more comprehensive health outcome perspective. Moreover, our method of estimating prescriptions using a probabilistic framework, particularly for the four conditions for which we associated drugs using DrugBank, is imperfect. There exists a possibility that we missed certain drug names, or that medications designed for alternate conditions could potentially be inaccurately included. This limitation of our study arises from our labeling method for prescribed drugs. Drugs are labeled according to associated conditions as sourced from the DrugBank database, without claiming any specific intent behind the prescription from the GP. While we can ascertain that a drug is likely prescribed for a given condition, it is worth noting that drugs can be associated with multiple conditions, both as per DrugBank, and in prescriptions by a GP. This multi-condition association increases the chances of co-prescriptions in our dataset. However, numerous studies on prescription patterns, from antihypertensive [34], to antidepressants [44] to anxiolytics [35] highlight the importance of prescriptions as a health outcome per se and its significance in the field of public health outcomes. We also emphasize that the initial drug list output by DrugBank can be augmented with human expert knowledge in a mixed-method approach to ensure the most accurate results. Our analyses show that the correlations between prescription prevalence scores derived with the automatic and manual methods range from .94 (for anxiety) to .99 for diabetes (see Appendix section D.4.3). Second, our dataset exclusively covers England, representing a single developed country. It is noteworthy that England provides high-quality data on both prescriptions and auxiliary census information, and the methods and insights derived from MEDSAT can serve as a foundation for the development of ML approaches that can subsequently be applied to developing countries once high-quality data becomes available, fostering progress towards tracking SDG about health. Third, there is the risk of stigmatizing certain communities based on our dataset. We believe that listed benefits and opportunities offered by MEDSAT outweigh this risk, and we invite ML and health research communities to employing ethical considerations, fostering inclusivity, and ensuring that the insights gained from MEDSAT are used to enact positive change, promote equity, and reduce health disparities.

## Acknowledgments and Disclosure of Funding

Project supported by AI4EO Beyond Fellowship at The International AI for Earth Observation Future Lab at TUM/DLR, ESA Network of Resources Initiative, German Federal Ministry for Economic Affairs and Climate Action in the framework of the "national center of excellence ML4Earth" (grant number: 50EE2201C), Munich Center for Machine Learning, Nokia Bell Labs, and European Union's

Horizon 2020 research and innovation programme under grant agreement No. 869764, awarded to the GoGreenRoutes project.

The authors also thank Sinziana Oncioiu, and Dario Augusto Borges Oliveira for helpful discussions, and Paolo Campanella and Bertrand Robert for their support during the dataset creation.

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

# Appendix
# MEDSAT: A Public Health Dataset for England Featuring Medical Prescriptions and Satellite Imagery

## Contents

## A  Dataset Licenses

The dataset is released under the CC BY-SA 4.0 license.

37th Conference on Neural Information Processing Systems (NeurIPS 2023) Track on Datasets and Benchmarks.

## B  Dataset Storage and Maintenance Plans

The dataset is available and will be maintained on TUMMedia, a data sharing service from Technical University Munich. It can be accessed via the following link: `https://doi.org/10.14459/2023mp1714817`. The code for datasets creation and experiments can be accessed on `https://github.com/sanja7s/MedSat`.

## C  Datasheet

| **Motivation** |
|---|

**For what purpose was the dataset created?** Was there a specific task in mind? Was there a specific gap that needed to be filled? Please provide a description.

Understanding the impact of the environment on human health is crucial for informing policy-making, promoting human well-being, and implementing timely health interventions, particularly in the context of increasing climate change-induced extreme weather events. Earth Observation data has emerged as a valuable resource for addressing critical challenges related to climate change, food security, and poverty [6, 15]. However, current applications that monitor population health using such data have limitations. Health indicators are often derived from surveys [46] and crowdsourcing [24], which may not be representative of the entire population. Additionally, these applications rely on a narrow set of environmental and sociodemographic indicators, hindering a comprehensive understanding of the relevant health factors [5, 17, 13]. Moreover, many of these datasets are not publicly available.

In this study, we aim to address these challenges by providing a publicly available dataset that harnesses the potential of earth observation data to monitor and comprehend the environmental influence on health outcomes. Our dataset combines four complementary data sources to enable comprehensive health modeling across the entire population of England. Specifically, for the years 2019 and 2020, we integrated medical prescription data at the practice level from the NHS, satellite-derived environmental features, Sentinel-2 satellite imagery, and sociodemographic indicators obtained from the latest UK census in 2021. These data sources were combined at the level of Lower Layer Super Output Areas (Lower Layer Super Output Area (LSOA)), which represent small administrative units ranging in size from 1km² to 18km². Our objectives are to:

- Conduct comprehensive health modeling by relating medical prescriptions to environmental, image, and sociodemographic features at a fine-grained spatial level.
- Identify relevant indicators for population health and uncover regional health disparities.
- Understand trends and factors influencing population health over time, including the impact of pandemic years on relevant health factors.

**Who created this dataset (e.g., which team, research group) and on behalf of which entity (e.g., company, institution, organization)?**

The dataset was created by researchers from:

- **Nokia Bell Labs**, Cambridge, UK.
- **Technical University of Munich**, Munich, Germany.
- **Wasdi platform** (https://www.wasdi.cloud/).

**Who funded the creation of the dataset?** If there is an associated grant, please provide the name of the grantor and the grant name and number.

The creation of the dataset was partly funded by the:

- Nokia Bell Labs,
- German Federal Ministry for Economic Affairs and Climate Action in the framework of the "national center of excellence ML4Earth" (grant number: 50EE2201C),

- Munich Center for Machine Learning,
- AI4EO Beyond Fellowship by The International AI for Earth Observation Future Lab at TUM/DLR,
- ESA Network of Resources Initiative, and
- European Union's Horizon 2020 research and innovation programme under grant agreement No. 869764, awarded to the GoGreenRoutes project.

---

**Composition**

---

### What do the instances that comprise the dataset represent (e.g., documents, photos, people, countries)? Are there multiple types of instances (e.g., movies, users, and ratings; people and interactions between them; nodes and edges)? Please provide a description.

For the years covered (2019 and 2020), our dataset contains the following two types of instances:

- **LSOA instance**: Our dataset covers the entire of England by including an instance per every LSOA in England. These instances are represented by point features describing the LSOAs environmental and sociodemographic indicators.
- **Sentinel-2 tile composite instance**: For the benefits of using computer vision to extract rich visual concepts that define environmental health, we provide a set of composite Sentinel-2 satellite image tiles per meteorological season covering the entire of England.

### How many instances are there in total (of each type, if appropriate)?

In each year, there are **33755 LSOA instances** (one per every LSOA in England) and $4 \times 35$ **composite tile instances** (one set of tiles per every meteorological season).

### Does the dataset contain all possible instances or is it a sample (not necessarily random) of instances from a larger set? If the dataset is a sample, then what is the larger set? Is the sample representative of the larger set (e.g., geographic coverage)? If so, please describe how this representativeness was validated/verified. If it is not representative of the larger set, please describe why not (e.g., to cover a more diverse range of instances, because instances were withheld or unavailable).

Our dataset provides a comprehensive overview of the health landscape in England by incorporating data from *all* LSOAs throughout the country. The prescription outcomes are derived from *all* prescribed items by the NHS, encompassing all patients within the nation during the specified years. Additionally, the census data covers the entire population. In terms of environmental variables, we computed yearly average scores using either all non-cloudy observations or a substantial sample of over 30% of images (e.g., for DynamicWorld) due to computational constraints. This extensive coverage allows for a thorough investigation of health-related factors at a national level. However, it is essential to acknowledge that the representativeness of our dataset diminishes when examining larger geographical regions due to the absence of data from developing countries, and the inherent specificity present even among developed countries.

### What data does each instance consist of? "Raw" data (e.g., unprocessed text or images) or features? In either case, please provide a description.

The **LSOA instances** are represented by feature vectors that contain environmental and sociodemographic indicators aggregated over the geographical area covered by an LSOA. In total, there are 43 environmental features describing air quality, greenery, climate, and land-cover distribution. Further, we include 111 sociodemographic variables that represent population counts, age group distribution, gender, ethnicity, religion, marital status, employment status, commuting habits, residence and housing, self-reported health, language skills, deprivation and income (the only non-census variable) levels. On the other hand, the **Sentinel-2 tile composite instances** contain 11 spectral bands that include aerosols, RGB bands, vegetation red edge bands, near-infrared bands and short-wave infrared bands. A detailed overview of the environmental and sociodemographic features and the Sentinel-2

image bands are provided in Figure 1 in the main manuscript and in the *"MedSat Variables.csv"* file in the public dataset directory.

**Is there a label or target associated with each instance?** If so, please provide a description.

The target variables in our dataset represent medical prescription prevalences on a yearly level. We included the following 7 target variables representing the quantity of:

- **diabetes** and **hypertension** prescriptions associated with *metabolic* conditions,
- **depression** and **anxiety** prescriptions associated with *mental* conditions,
- **asthma** prescriptions associated with *respiratory* conditions,
- **opioids** prescriptions which are predominantly prescribed for pain management, and
- **total** prescriptions as a proxy for general health and well-being.

**Is any information missing from individual instances?** If so, please provide a description, explaining why this information is missing (e.g., because it was unavailable). This does not include intentionally removed information, but might include, e.g., redacted text.

In both years, 5,163 LSOA instances ($\sim$15% of the available instances) have missing feature values. The missing values for environmental features occur because some of the used satellite products do not provide these features for every LSOA. For example, there are 1,479 (4%) LSOAs for which the variables related to temperature, snow, or radiation derived from the European Centre for Medium-Range Weather Forecasts (ECMWF) Reanalysis v5 product [37] are missing. We observed that this product does not provide these features for LSOAs located near coastal regions. Next, there are 1,507 (4%) LSOAs for which information about the aerosol optical depth and PM2.5 particles is missing from the Copernicus Atmosphere Monitoring Service (CAMS) product [9]. In the future work, we aim to evaluate different satellite products that can provide the missing values for these features. There are 1,956 (6%) LSOAs with missing outcome data. This absence doesn't stem from a lack of prescription data from the NHS. Instead, it arises from a discrepancy between the LSOA shapefiles of the patient data used for outcome calculation from 2018 and the shapefiles from the 2021 census. We opted to utilize the most recent 2021 shapefiles. We await forthcoming releases of NHS patient data that align with these shapefiles, allowing us to incorporate the data for the missing LSOAs. For the sociodemographic indicators, we found that the net annual income is missing for 537 LSOAs as this data was not provided on an LSOA level in the UK census but we derived from the latest 2018 estimates of mean annual household income for Middle layer Super Output Areas (MSOAs) from the Family Resources Survey (and there is a mismatch for a small number of LSOAs and MSOAs due to the latest adjustments to the LSOAs boundaries in 2021).

Notably, we possess Census sociodemographic and image features data for all LSOAs. Depending on the specific analyses intended, the maximum missing value rate could be constrained by the outcomes, standing at 5.7%. For instance, this applies if a user does not require environmental variables from ECMWF or CAMS products.

**Are relationships between individual instances made explicit (e.g., users' movie ratings, social network links)?** If so, please describe how these relationships are made explicit.

To combine both types of instances, we provide shapefiles describing the LSOAs coordinates that allow extraction of image features for individual LSOAs from the composite Sentinel-2 satellite image. Based on these shapefiles, in our code repository we provide an implementation for the extraction of LSOA-specific image features out of the Sentinel-2 images. Further, the region column in the spatial data files determines the larger geographical area to which the LSOA belongs and can be used to cluster the LSOAs based on their spatial proximity and to perform health analysis over a larger geographical area.

**Are there recommended data splits (e.g., training, development/validation, testing)?** If so, please provide a description of these splits, explaining the rationale behind them.

We recommend the spatial data split described in Section E.2.2 where LSOAs are first clustered into blocks of sizes 28km x 28km. Next, those blocks are randomly assigned to 5 folds, each fold

containing an equal number of blocks. These procedure ensures that LSOAs belonging to a same block do not appear within the training and the test set.

**Are there any errors, sources of noise, or redundancies in the dataset?** If so, please provide a description.

The environmental and image features utilized in our dataset are derived from high-level satellite products, which undergo preprocessing to mitigate common errors associated with remote sensing acquisition. However, it is important to note that our subsequent processing and spatial aggregation of this data may introduce sources of noise. For instance, we employ a threshold of 0.2 for NDVI values to calculate the fraction of greenery pixels per LSOA. Consequently, different threshold values may yield varying results. Additionally, the prescription values are derived through probabilistic associations between patients and nearby LSOAs, as well as drug names and corresponding conditions. This process, while valuable, is not without imperfections and introduces some level of noise.

In terms of redundancies, our dataset contains many correlated features among the environmental and sociodemographic indicators. A subset of correlated environmental features is presented in Figure 1 where it can be seen that the temperature feature is strongly positively correlated with surface soil temperature, thermal radiation, and atmospheric features while being strongly negatively correlated with snow density. These correlations represent a challenge for many machine learning models and we encourage the researches to investigate approaches for feature selection that can lead to more efficient and accurate health modeling.

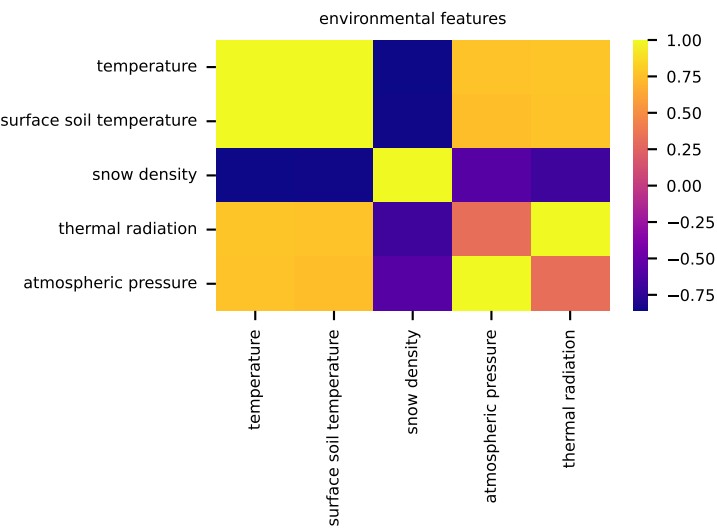

Figure 1: Example of correlated environmental features.

**Is the dataset self-contained, or does it link to or otherwise rely on external resources (e.g., websites, tweets, other datasets)?** If it links to or relies on external resources, a) are there guarantees that they will exist, and remain constant, over time; b) are there official archival versions of the complete dataset (i.e., including the external resources as they existed at the time the dataset was created); c) are there any restrictions (e.g., licenses, fees) associated with any of the external resources that might apply to a future user? Please provide descriptions of all external resources and any restrictions associated with them, as well as links or other access points, as appropriate.

The dataset is self-contained.

**Does the dataset contain data that might be considered confidential (e.g., data that is protected by legal privilege or by doctor-patient confidentiality, data that includes the content of individuals non-public communications)?** If so, please provide a description.

Our four complementary dataset components are derived from public data sources. As such, they are not confidential.

**Does the dataset contain data that, if viewed directly, might be offensive, insulting, threatening, or might otherwise cause anxiety?** If so, please describe why.

Our dataset does not contain any data that can be directly perceived as offensive or inappropriate. As mentioned in the limitations section, the only potential concern is the possibility of stigmatization at the level of LSOA. However, it is worth noting that such stigmatization can also be derived from other publicly available data sources.

**Does the dataset relate to people?** If not, you may skip the remaining questions in this section.

Yes.

**Does the dataset identify any subpopulations (e.g., by age, gender)?** If so, please describe how these subpopulations are identified and provide a description of their respective distributions within the dataset.

Our dataset identifies a percentage of subpopulations by age, ethnicity, and gender across LSOAs derived from the UK census. However, these features do not point to any personally identifiable information because the census implements stringent privacy protection measures, including targeted record swapping and cell key perturbation, to ensure data confidentiality without compromising aggregated statistics [1]. The age group and ethnicity distributions are visualized in Figure 2 which shows that the age groups are balanced except for the age groups over 80 years which occur less frequently compared to the age other groups. When it comes to ethnicity and gender, the White ethnicity is dominant in the dataset while the gender distribution is balanced.

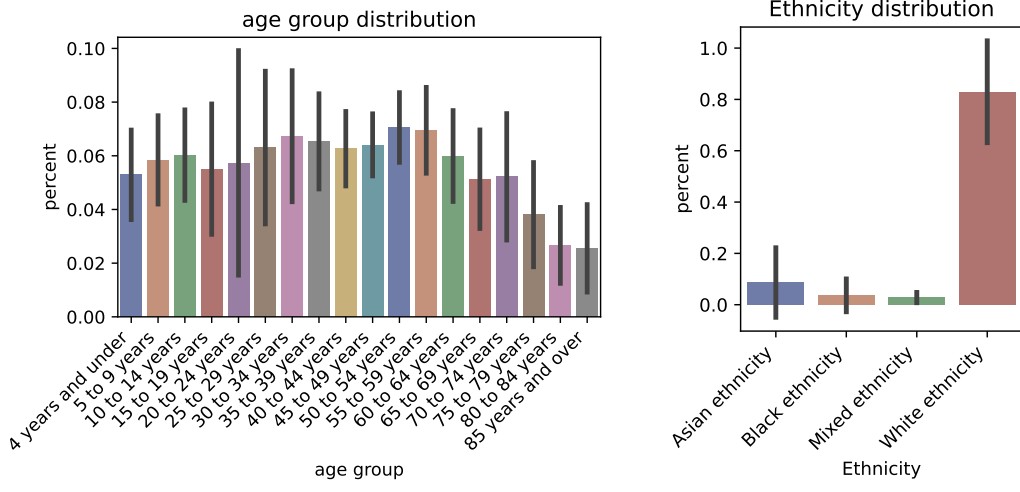

Figure 2: Age group and ethnicity distributions

**Is it possible to identify individuals (i.e., one or more natural persons), either directly or indirectly (i.e., in combination with other data) from the dataset?** If so, please describe how.

No, for the reasons stated above.

**Does the dataset contain data that might be considered sensitive in any way (e.g., data that reveals racial or ethnic origins, sexual orientations, religious beliefs, political opinions or union memberships, or locations; financial or health data; biometric or genetic data; forms of government identification, such as social security numbers; criminal history)?** If so, please provide a description.

Certainly, we have extracted sensitive information such as ethnicity, religious beliefs, and economic deprivation per LSOA from the census data. However, it is important to note that this information is publicly available through census records.

---

**Collection Process**

---

**How was the data associated with each instance acquired?** Was the data directly observable (e.g., raw text, movie ratings), reported by subjects (e.g., survey responses), or indirectly inferred/derived from other data (e.g., part-of-speech tags, model-based guesses for age or language)? If data was reported by subjects or indirectly inferred/derived from other data, was the data validated/verified? If so, please describe how.

The environmental variables were already preprocessed by the satellite products, and we didn't perform further validation. They have varying frequencies, from daily to monthly, depending on the specific product. In our dataset, we created the environmental features by averaging the variables derived from these products on a yearly level. The Sentinel-2 tiles were derived by aggregating publicly available, cloud-free Sentinel-2 images per calendar season that were already preprocessed by the Sentinel-2 mission. Further, the sociodemographic variables are obtained as single values from the UK census data, and as such, do not require validation. The target variables include yearly aggregates of prescription quantities issued by the NHS health authorities. While there is limited condition prevalence data available at the LSOA level across England, our tests for London at the ward level showed moderate to high correlations between our prescription prevalence scores and condition prevalence, ranging from 0.76 for diabetes to 0.89 for depression. However, it is important to note that prescription scores and condition prevalence may not correlate perfectly, as they capture different health constructs.

**What mechanisms or procedures were used to collect the data (e.g., hardware apparatus or sensor, manual human curation, software program, software API)?** How were these mechanisms or procedures validated?

In our data collection process, we developed our own parser for extracting prescription data. Additionally, for gathering environmental variables, we utilized the Google Earth Engine platform [14] for point data collection and the WASDI platform [44] for image data collection. The code for data collection and preprocessing is available on our GitHub repository.

**If the dataset is a sample from a larger set, what was the sampling strategy (e.g., deterministic, probabilistic with specific sampling probabilities)?**

NA

**Who was involved in the data collection process (e.g., students, crowdworkers, contractors) and how were they compensated (e.g., how much were crowdworkers paid)?**

The data collection process for this study was carried out solely by the co-authors, who were involved in the research as part of their respective roles.

**Over what timeframe was the data collected? Does this timeframe match the creation timeframe of the data associated with the instances (e.g., recent crawl of old news articles)?** If not, please describe the timeframe in which the data associated with the instances was created.

All the data were collected in 2022.

**Were any ethical review processes conducted (e.g., by an institutional review board)?** If so, please provide a description of these review processes, including the outcomes, as well as a link or other access point to any supporting documentation.

As publicly available data was used, this was not needed.

**Does the dataset relate to people?** If not, you may skip the remaining questions in this section.

No.

**Did you collect the data from the individuals in question directly, or obtain it via third parties or other sources (e.g., websites)?**

NA

**Were the individuals in question notified about the data collection?** If so, please describe (or show with screenshots or other information) how notice was provided, and provide a link or other access point to, or otherwise reproduce, the exact language of the notification itself.

NA

**Did the individuals in question consent to the collection and use of their data?** If so, please describe (or show with screenshots or other information) how consent was requested and provided, and provide a link or other access point to, or otherwise reproduce, the exact language to which the individuals consented.

NA

**If consent was obtained, were the consenting individuals provided with a mechanism to revoke their consent in the future or for certain uses?** If so, please provide a description, as well as a link or other access point to the mechanism (if appropriate).

NA

**Has an analysis of the potential impact of the dataset and its use on data subjects (e.g., a data protection impact analysis) been conducted?** If so, please provide a description of this analysis, including the outcomes, as well as a link or other access point to any supporting documentation.

NA

| Preprocessing/cleaning/labeling |
|:---:|

**Was any preprocessing/cleaning/labeling of the data done (e.g., discretization or bucketing, tokenization, part-of-speech tagging, SIFT feature extraction, removal of instances, processing of missing values)?** If so, please provide a description. If not, you may skip the remainder of the questions in this section.

Yes. The environmental features were averaged on a yearly level, the prescription quantities were normalized according to the number of patients residing in an LSOA, and the Sentinel-2 images were averaged per calendar season.

**Was the "raw" data saved in addition to the preprocessed/cleaned/labeled data (e.g., to support unanticipated future uses)?** If so, please provide a link or other access point to the "raw" data.

No, but the used data sources are available publicly from the NHS and ONS wesbites, and GEE and WASDI platforms.

**Is the software used to preprocess/clean/label the instances available?** If so, please provide a link or other access point.

Yes, on the provided github repository.

| Uses |
|:---:|

**Has the dataset been used for any tasks already?** If so, please provide a description.

In the main manuscript, we have presented the potential value of our dataset in the context of prescription prediction, analysis of relevant health factors, and examination of health disparities. However, there have been no published works utilizing this dataset to date.

**Is there a repository that links to any or all papers or systems that use the dataset?** If so, please provide a link or other access point.

NA.

**What (other) tasks could the dataset be used for?**

The geographical division within our dataset presents a challenge in developing machine learning models that can effectively generalize health findings across diverse regions. Additionally, the inclusion of two years of data in our dataset offers the potential for predicting future health outcomes based on historical health data.

**Is there anything about the composition of the dataset or the way it was collected and preprocessed/cleaned/labeled that might impact future uses?** For example, is there anything that a future user might need to know to avoid uses that could result in unfair treatment of individuals or groups (e.g., stereotyping, quality of service issues) or other undesirable harms (e.g., financial harms, legal risks) If so, please provide a description. Is there anything a future user could do to mitigate these undesirable harms?

We have meticulously collected and preprocessed the data in accordance with the highest ethical standards to prevent any misuse of our dataset. We strongly urge fellow researchers to conduct responsible analyses and utilize the dataset with integrity.

**Are there tasks for which the dataset should not be used?** If so, please provide a description.

We caution against presenting the results from our dataset in any stigmatizing way, such as highlighting the worst areas based on certain attributes, without a clear purpose or intention to address the underlying issues. It is important to use the data responsibly and consider the potential implications of how the results are presented and interpreted.

---

| Distribution |
|:---:|

**Will the dataset be distributed to third parties outside of the entity (e.g., company, institution, organization) on behalf of which the dataset was created?** If so, please provide a description.

The dataset will be publicly available.

**How will the dataset will be distributed (e.g., tarball on website, API, GitHub)** Does the dataset have a digital object identifier (DOI)?

As stated in Section B, the dataset is publicly available on TUMMedia through the following DOI: `https://doi.org/10.14459/2023mp1714817`

**When will the dataset be distributed?**

The dataset is publicly available.

**Will the dataset be distributed under a copyright or other intellectual property (IP) license, and/or under applicable terms of use (ToU)?** If so, please describe this license and/or ToU, and provide a link or other access point to, or otherwise reproduce, any relevant licensing terms or ToU, as well as any fees associated with these restrictions.

No.

**Have any third parties imposed IP-based or other restrictions on the data associated with the instances?** If so, please describe these restrictions, and provide a link or other access point to, or otherwise reproduce, any relevant licensing terms, as well as any fees associated with these restrictions.

No.

**Do any export controls or other regulatory restrictions apply to the dataset or to individual instances?** If so, please describe these restrictions, and provide a link or other access point to, or otherwise reproduce, any supporting documentation.

No.

| Maintenance |
|:---:|

**Who will be supporting/hosting/maintaining the dataset?**

TUMMedia will host and maintain the dataset which ensures the long-term accessibility and citability of our dataset, facilitating its use and reference by the research community.

**How can the owner/curator/manager of the dataset be contacted (e.g., email address)?**

The email addresses of the authors will be available on the GitHub page.

**Is there an erratum?** If so, please provide a link or other access point.

No.

**Will the dataset be updated (e.g., to correct labeling errors, add new instances, delete instances)?** If so, please describe how often, by whom, and how updates will be communicated to users (e.g., mailing list, GitHub)?

As part of our ongoing efforts, we are currently developing a prescription parser that enables the calculation of prescription quantities on a monthly basis and going up to 10 years in the past. Additionally, we are enhancing the drug matching module by leveraging state-of-the-art Large Language Models (LLMs). Once these tasks are completed, we will be able to update the GitHub with the updated parser so that interested researchers can calculated additional target variables and enhance the temporal granularity of those features.

**If the dataset relates to people, are there applicable limits on the retention of the data associated with the instances (e.g., were individuals in question told that their data would be retained for a fixed period of time and then deleted)?** If so, please describe these limits and explain how they will be enforced.

No.

**Will older versions of the dataset continue to be supported/hosted/maintained?** If so, please describe how. If not, please describe how its obsolescence will be communicated to users.

No.

**If others want to extend/augment/build on/contribute to the dataset, is there a mechanism for them to do so?** If so, please provide a description. Will these contributions be validated/verified? If so, please describe how. If not, why not? Is there a process for communicating/distributing these contributions to other users? If so, please provide a description.

Yes, the code for collecting the four complementary components of the MEDSAT datasets is publicly available on our GitHub repository: `https://github.com/sanja7s/MedSat`.

# D  Dataset Details

## D.1  Sociodemographic Features

All the sociodemographic variables are collected from the UK 2021 Census using `https://www.nomisweb.co.uk/sources/census_2021`, except for `income`, which is collected from Family Resources Survey 2018 `https://www.gov.uk/government/collections/family-resources-survey--2`.

## D.2  Environmental Features

For this release of the MEDSAT dataset, we used the sources listed in Table 1 to derive our environmental features. We performed spatial reduction and averaging of yearly values at the LSOA level. When reducing image data to our areas of interest, we employed specific spatial resolution scaling values: 1000 m for Sentinel-5, CAMS, and TOMS&OMI, and 10 m for ESA-WorldCover and Sentinel-2. These choices were guided by the original products' resolutions and the processing limitations within the GEE platform.

Table 1: Sources and Statistics about of Environmental Variables. The links below should all be prepended with `https://developers.google.com/earth-engine/datasets/catalog/`

| Type | Satellite/Source | Scale (m) | Resolution (m) | Link to data source |
|------|------------------|-----------|----------------|---------------------|
| Air Q | Sentinel-5 | 1000 | 1113.2 | `COPERNICUS_S5P_OFFL_L3_NO2` |
| Air Q | TOMS&OMI | 1000 | 111000 | `TOMS_MERGED` |
| Air Q | CAMS | 5000 | 44528 | `ECMWF_CAMS_NRT` |
| greenery | Sentinel-2 | 50 | 10 | `COPERNICUS_S2_SR` |
| greenery | ERA5-ECMWF | 100 | 11132 | `ECMWF_ERA5_LAND_MONTHLY_AGGR` |
| climate | ERA5-ECMWF | 100 | 11132 | `ECMWF_ERA5_LAND_MONTHLY_AGGR` |
| land cover | DynamicWorld | 200 | 10 | `GOOGLE_DYNAMICWORLD_V1` |

## D.3  Image Features

To generate composite images, we obtained raw Sentinel-2 satellite data and processed it using the WASDI platform (`https://www.wasdi.net`), which allows access and online processing of both public and commercial datasets. For our analysis, we processed Sentinel-2 images corresponding to the years 2019 and 2020 by calculating the average values for each of the four meteorological seasons in the both years, as defined by the Met Office [32]. Each Sentinel-2 image consists of 13 bands, with four bands at a spatial resolution of 10 m, six bands at 20 m, and three bands at 60 m. We focused on 11 specific bands, as described in main manuscript, which capture environmental factors relevant to health (e.g., B01 captures aerosols, representing air quality). We excluded bands B09 and B10, as the former is primarily used for cirrus cloud detection and the latter is used to map water vapor.

To generate composite images for each season, we collected and parsed all images within the corresponding three-month period (on average $\sim 1000$ images per season). This amounted to parsing around 2TB per season and around 8TB per year (see Table 2). For image processing, we conducted several steps to ensure data consistency and quality. Firstly, we resampled the selected bands from their original resolutions of 60 m and 20 m to a uniform resolution of 10 m. This resampling allowed for a consistent analysis across all bands. Next, we applied the cloud mask provided with each Sentinel-2 image to each of the 11 bands. This cloud mask effectively identified and excluded pixels affected by cloud cover, ensuring the accuracy and reliability of the data. Then, we computed a pixel-per-pixel average over time, considering only values observed in cloud-free conditions. This averaging process ensured that the final composite images represented the typical environmental characteristics for each season. Visualizations of a subset of composite bands for the summer of 2020 for LSOAs with the highest and lowest total prescriptions per capita are shown in Figure 3.

### D.3.1  Extracting Image Features per LSOA

To extract per-LSOA features from Sentinel-2 composite images, we segmented each image into LSOA-specific imagelets with the procedure depicted in Figure 4. For every imagelet, we computed five descriptive statistics: *min, max, mean, std*, and *median*. Our examination of these statistics

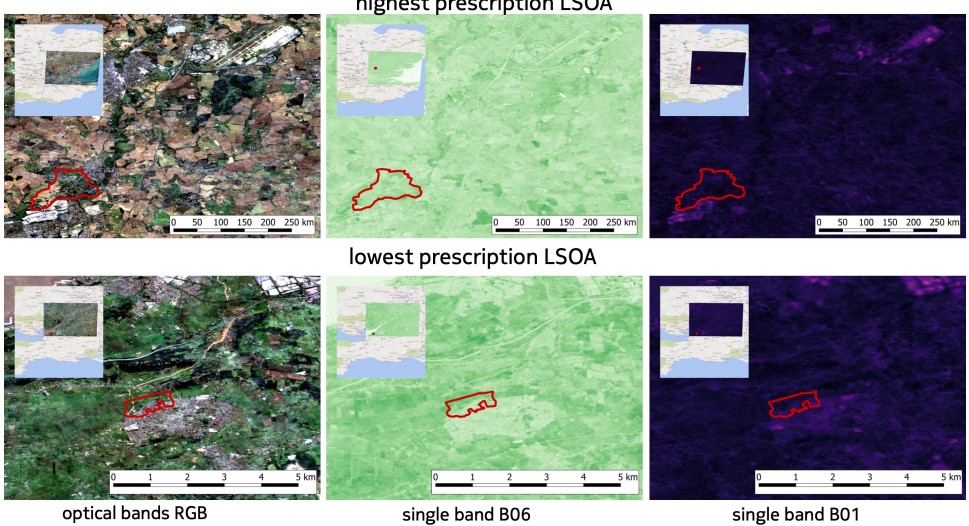

highest prescription LSOA

lowest prescription LSOA

optical bands RGB      single band B06      single band B01

Figure 3: **Visualization of the selected MEDSAT Sentinel-2 composite bands for the LSOAs with highest/lowest total prescriptions quantity per capita.**

revealed inter-correlations, leading to three distinct classes of correlated bands for each image (e.g., cross-correlation of mean values between summer and winter composite images for 2020 can be seen in Figure 5). Further assessment of the statistics' distributions showed skewness in some (like *min*) and both skewness and extreme outliers in others (specifically *max*). Moreover, certain statistics, such as *mean* and *median*, exhibited inherent correlations.

### D.4 Prescription Outcomes

#### D.4.1 Existing Public and Population Health Datasets

Fine-grained spatial and temporal indices for prevalence of medical conditions are rarely available. Public health agencies typically collect data infrequently through representative surveys (e.g., National health and nutrition examination survey (NHANES) [11], Health Survey for England (HSE) [29], or Behavioral Risk Factor Surveillance System (BRFSS) [33]) or on population samples with cohort studies (e.g., Framingham Heart Study [26], The Swiss National Cohort (SNC) [38], or The UK Biobank [39]). Surveys come with well-known biases, such as sampling [7], non-response [36], recall [19], or question wording [40] bias. While cohort studies aim at limiting these biases by complementing participant questionnaire responses with their health records, or even genetic

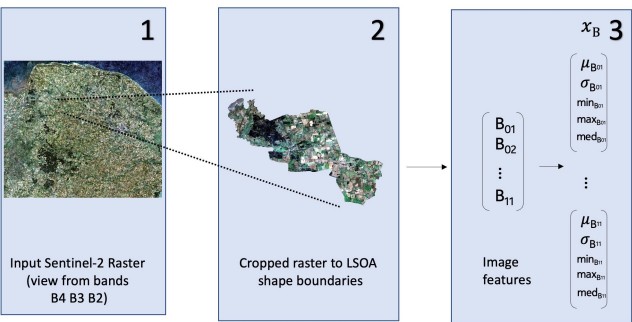

Figure 4: **Extracting Image Features:** For each seasonal Sentinel-2 composite image, we extracted a set of five metrics per LSOA from each band $B_i \in \{B_{01}, \ldots, B_{11}\}$: *mean, stdev, min, max,* and *median*. While we show only 3 representative bands for visualization purposes, features were extracted from all 11 bands, resulting in a total of 55 features per seasonal image.

Table 2: The size of the data processed on WASDI platform while producing the environmental image features for each month.

| date from | date to | GB |
|---|---|---|
| 2018-12-01 | 2018-12-31 | 671.4277637 |
| 2019-01-01 | 2019-01-31 | 671.1432031 |
| 2019-02-01 | 2019-02-31 | 652.6567187 |
| 2019-03-01 | 2019-03-31 | 648.4059570 |
| 2019-04-01 | 2019-04-31 | 632.4159082 |
| 2019-05-01 | 2019-05-31 | 617.2984863 |
| 2019-06-01 | 2019-06-31 | 641.2775684 |
| 2019-07-01 | 2019-07-31 | 650.5995117 |
| 2019-08-01 | 2019-08-31 | 661.7470703 |
| 2019-09-01 | 2019-09-31 | 631.0063281 |
| 2019-10-01 | 2019-10-31 | 631.5184570 |
| 2019-11-01 | 2019-11-31 | 666.4724512 |
| 2019-12-01 | 2019-12-31 | 688.0771484 |
| 2020-01-01 | 2020-01-31 | 657.6159668 |
| 2020-02-01 | 2020-02-31 | 648.8922559 |
| 2020-03-01 | 2020-03-31 | 614.7591406 |
| 2020-04-01 | 2020-04-31 | 654.5712695 |
| 2020-05-01 | 2020-05-31 | 676.6889844 |
| 2020-06-01 | 2020-06-31 | 640.9450684 |
| 2020-07-01 | 2020-07-31 | 648.6113184 |
| 2020-08-01 | 2020-08-31 | 647.2279785 |
| 2020-09-01 | 2020-09-31 | 654.1471207 |
| 2020-10-01 | 2020-10-31 | 648.9410100 |
| 2020-11-01 | 2020-11-31 | 655.3504297 |
| 2020-12-01 | 2020-12-31 | 622.1216504 |
| 2021-01-01 | 2021-01-31 | 672.5485547 |

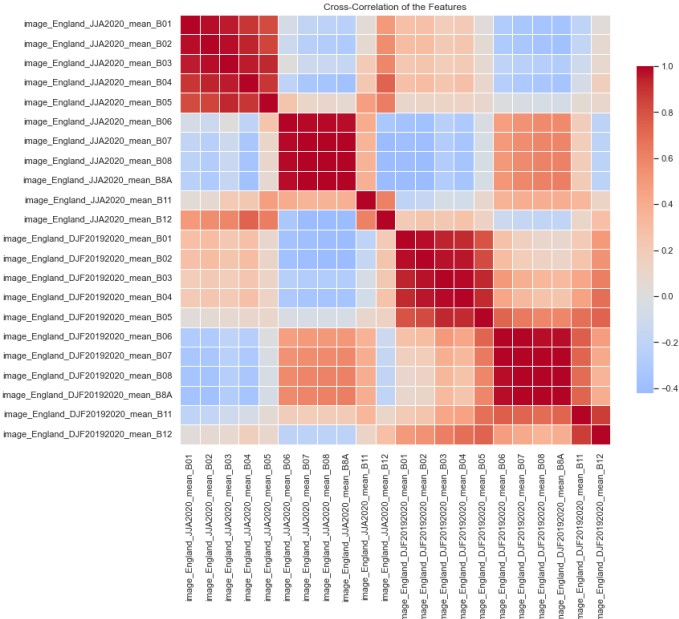

Figure 5: **Cross-correlation matrix for mean band values extracted from summer (JJA) and winter (DJF) Sentinel-2 composites for 2020.** We see that bands $B01 - B05$, and $B06 - B8A$, as well as $B11 - B12$ form clusters of highly correlated features.

information, they are still limited in size, can be expensive and time-consuming to conduct, and participants may drop out of the study. The All of US Research Program (AoURP) [34] is a unique effort in the US recruiting over 1 million participants to study precision population health. However,

despite its size, AoURP still might face limited representativeness, and the success of the program will depend on long-term engagement and retention of participants. In summary, exhaustive health outcomes data are often of limited scope and granularity, and are prone to a variety of biases.

### D.4.2   NHS Medical Prescriptions

The practice-level prescribing data in England has been published monthly by the National Health Services (NHS) Business Services Authority since 1998. This data provides information on the quantity and cost of prescribed medications for each *general practitioner (GP) practice* in the country. Although freely accessible, it is provided in individual files for each year, and is difficult to combine in order to analyze spatial or temporal trends. The OpenPrescribing service [10] enables the exploration of trends in items, cost, price per item, and quantity per item for each medical prescription and GP practice from 1998. However, a single GP practice can cater to patients from multiple nearby administrative areas (e.g., LSOAs). Consequently, this service does not directly provide insights into the spatial prevalence of prescriptions across these areas. The PrAna R package [21] allows the calculation of prescribed quantities of active pharmaceutical ingredients (APIs) by postcode. However, a single API or medical drug can be used to treat various conditions (e.g., selective serotonin reuptake inhibitors (SSRIs) are used for treating depression, anxiety disorders, and certain eating disorders), and a specific medical condition is typically treated using multiple drugs or APIs (e.g., depression is treated with both SSRIs and tricyclic antidepressants (TCAs)). This limits the ability of such a tool to understand the prevalence of prescriptions for specific conditions. As a result, previous studies using prescribing data in England have mainly focused on a small number of drugs manually aggregated for bespoke analyses, primarily examining temporal trends (e.g., [20, 30, 8, 45, 12]), with a few studies investigating also spatial trends (e.g., [35, 18]). Despite the valuable insights provided by previous studies using prescribing data, to the best of our knowledge, there is currently no approach that generalizes across various medical conditions to calculate spatial and temporal trends from this data.

**Original NHS Practice-Level Prescribing Data**   The monthly practice-level prescribing data in England, provided by the National Health Services (NHS) since July 2010 [31], constitutes the foundation of our analysis. The dataset consists of four files visualized in Figure 6 containing the following information:

1. *GP monthly prescriptions* – Anonymized prescriptions across General Practitioner (GP) practices in England for a given month. Each prescription entry includes details such as the drug name, British National Formulary (BNF) code [31], practice code, total number of items, total cost, and individual item quantities.

2. *Drugs* – A comprehensive list of drugs with their unique BNF codes [31].

3. *GPs* – Information on GP practices, including practice codes, names, and full addresses. Our dataset encompasses 6,924 GPs located across England, excluding closed or prison-hosted practices.

4. *Patients* – Contains practice codes, census Lower-layer Super Output Area (LSOA) codes, and the number of patients registered with each practice in a particular area. By aggregating this data, we calculated the total number of primary care patients residing in a specific area using the equation:

$$n_{\text{pat}}(a) = \sum_{\text{gp}} n_{\text{pat}}(\text{gp}, a) \tag{1}$$

Here, $n_{\text{pat}}(\text{gp}, a)$ represents the number of patients registered with a particular GP residing in area $a$. The strong correlation ($r = .92$) between the number of patients and residents in an LSOA validates our probabilistic approach of assigning patients to areas. Additionally, we computed the fraction of a GP practice's patients associated with a specific area $a$ using the equation:

$$f(\text{gp}, a) = \frac{n_{\text{pat}}(\text{gp}, a)}{n_{\text{pat}}(\text{gp})} \tag{2}$$

| Practice Code | BNF Code | Drug Name | Items | Cost | Pills |
|---|---|---|---|---|---|
| ... | ... | ... | ... | ... | ... |
| N81013 | 0103010T0BBAAAA | Zantac_Tab_150 mg | 1.0 | 1.30 | 60 |
| ... | ... | ... | ... | ... | ... |

(a) NHS GP monthly prescriptions

| BNF Code | Drug |
|---|---|
| ... | ... |
| 0103010T0 | Ranitidine |
| ... | ... |

(b) NHS drugs

| Practice Code | Name | Postcode |
|---|---|---|
| ... | ... | ... |
| N81013 | High Street Surgery | SK11 6JL |
| ... | ... | ... |

(c) NHS GPs

| Practice Code | LSOA | # Patients |
|---|---|---|
| ... | ... | ... |
| N81013 | E01012198 | 10 |
| ... | ... | ... |

(d) NHS GP patients

Figure 6: The four NHS Datasets: *(a)* GP (general practitioner) monthly prescriptions; *(b)* Drugs; *(c)* GPs; and *(d)* Patients. Each monthly prescription in dataset *(a)* was translated into a drug name based on the BNF code for which dataset *(b)* offered the corresponding drug (preparation name). The prescription was also geographically mapped using the GP code for which dataset *(c)* provided the location. To then map the prescription at the level of census LSOAs, we computed the fraction of the GP's patients who lived in each LSOA from dataset *(d)*.

### D.4.3 DrugBank Network

We show here the lists of drug names extracted automatically from DrugBank for two conditions, i.e., diabetes (Table 3) and anxiety (Table 5), and the pre-existing list from the literature [16] for depression (Table 4). For anxiety, we also visualize the corresponding DrugBank network subset (Figure 7). For the other lists of drugs, please refer to our GitHub repository: `https://github.com/sanja7s/MedSat/tree/master/code/collate_data/NHS_prescription_parser/drug_names`.

The correlation between prevalence scores based on the list of drug names extracted automatically from DrugBank and based on the pre-existing list from the literature are .94 for anxiety and .99 for diabetes, attesting to the high quality of automatically generated output from DrugBank.

Table 3: List of drug names associated with **diabetes**.

| BNF Code | Drug name | BNF Code | Drug name |
|---|---|---|---|
| 0601023A0 | Acarbose | 0601023AM | Canagliflozin |
| 0601023AS | Albiglutide | 0601023AP | Canagliflozin/Metformin |
| 0601023AK | Alogliptin | 0601021E0 | Chlorpropamide |
| 0601023AJ | Alogliptin/Metformin | 0212000AD | Colesevelam Hydrochloride |
| 0607010B0 | Bromocriptine | 0601023AG | Dapagliflozin |
| 0601023AL | Dapagliflozin/Metformin | 0605020E0 | Desmopressin Acetate |
| 0601023AQ | Dulaglutide | 0601023AN | Empagliflozin |
| 0601023AR | Empagliflozin/Metformin | 0601023AX | Ertugliflozin |
| 0601023Y0 | Exenatide | 0601021H0 | Glibenclamide |
| 0601021M0 | Gliclazide | 0601021A0 | Glimepiride |
| 0601021P0 | Glipizide | 0202010L0 | Hydrochlorothiazide |
| 0202080M0 | Hydrochlorothiazide/Potassium | 0601011A0 | Insulin Aspart |
| 0601012Z0 | Insulin Degludec | 0601012X0 | Insulin Detemir |
| 0601012V0 | Insulin Glargine | 0601012AB | Insulin Glargine/Lixisenatide |
| 0601011P0 | Insulin Glulisine | 0601011R0 | Insulin Human |
| 0601011L0 | Insulin Lispro | 0601012S0 | Isophane Insulin |
| 0601023AE | Linagliptin | 0601023AF | Linagliptin/Metformin |
| 0601023AB | Liraglutide | 0601023AI | Lixisenatide |
| 0601022B0 | Metformin Hydrochloride | 0601023W0 | Metformin Hydrochloride/Pioglitazone |
| 0601023V0 | Metformin Hydrochloride/Rosiglitazone | 0601023AD | Metformin Hydrochloride/Sitagliptin |
| 0601023Z0 | Metformin Hydrochloride/Vildagliptin | 0601023M0 | Miglitol |
| 0601023U0 | Nateglinide | 0601023B0 | Pioglitazone Hydrochloride |
| 0601023R0 | Repaglinide | 0601023S0 | Rosiglitazone |
| 0601023AC | Saxagliptin | 0601023AV | Saxagliptin/Dapagliflozin |
| 0601023AH | Saxagliptin/Metformin | 0601023AW | Semaglutide |
| 0601023X0 | Sitagliptin | 0601021V0 | Tolazamide |
| 0601021X0 | Tolbutamide | 0601023AA | Vildagliptin |

Table 4: List of drug names associated with **depression** (source [16]).

| BNF Code | Drug name | BNF Code | Drug name |
|----------|-----------|----------|-----------|
| 0403030Q0 | Sertraline | 0403040T0 | Reboxetine |
| 0403030D0 | Citalopram | 0403040Z0 | Agomelatine |
| 0403030E0 | Fluoxetine | 0402010S0 | Flupentixol |
| 0403030P0 | Paroxetine | 0403010T0 | Tryptophan |
| 0403030X0 | Escitalopram | 0403040N0 | Nefazodone |
| 0403030F0 | Fluvoxamine | 0403010U0 | Oxitriptan |
| 0403040M0 | Mirtazapine | 0403010B0 | Amitriptyline |
| 0403040W0 | Venlafaxine | 0403010V0 | Trazodone |
| 0403040Y0 | Duloxetine | 0403010H0 | Dosulepin |
| 0403040AB | Vortioxetine | 0403010L0 | Lofepramine |
| 0403010R0 | Nortriptyline | 0403010C0 | Clomipramine |
| 0403010J0 | Imipramine | 0403010W0 | Trimipramine |
| 0403010G0 | Doxepin | 0403010M0 | Mianserin |
| 0403010A0 | Amoxapine | 0403010P0 | Moclobemide |
| 0403010X0 | Tranylcypromine | 0403010S0 | Phenelzine |
| 0403010K0 | Isocarboxazid | | |

Table 5: List of drug names associated with **anxiety**.

| BNF Code | Drug name | BNF Code | Drug name |
|----------|-----------|----------|-----------|
| 0401020A0 | Alprazolam | 0401020G0 | Bromazepam |
| 0403010B0 | Amitriptyline Hydrochloride | 0401020B0 | Buspirone Hydrochloride |
| 0401020D0 | Chlordiazepoxide | 0401020E0 | Chlordiazepoxide Hydrochloride |
| 0403030D0 | Citalopram Hydrobromide | 0401020V0 | Clorazepate Dipotassium |
| 0401020K0 | Diazepam | 0704020AA | Duloxetine Hydrochloride |
| 0403030X0 | Escitalopram | 0408010G0 | Gabapentin |
| 0304010J0 | Hydroxyzine Hydrochloride | 0401020P0 | Lorazepam |
| 0401020R0 | Meprobamate | 0403040X0 | Mirtazapine |
| 0401020T0 | Oxazepam | 0403030P0 | Paroxetine Hydrochloride |
| 0402010Q0 | Perphenazine | 0406000T0 | Prochlorperazine Maleate |
| 0204000R0 | Propranolol Hydrochloride | 0402010AB | Quetiapine |
| 0402010X0 | Trifluoperazine | 0403040W0 | Venlafaxine |

# E  Benchmarks Results

## E.1  Detailed Health Inequalities

In Figure 8, we present the MEDSAT data, providing insights into healthcare accessibility disparities across regions. Our analysis focuses on comparing the total number of registered patients with the population at the LSOA level, revealing significant spatial deviations, indicative of inequalities.

First, interestingly, we find a prevailing pattern where the number of registered patients exceeds the census population in most areas of the country. This aligns with previous investigations by UK authorities [42]. Second, although the correlation ($r = .87$, $p \approx 0$) is strong, certain LSOAs exhibit disproportionate patient-to-population ratios. For instance, Forest of Dean and Shropshire, located near the Welsh border, show lower patient numbers, likely due to residents being registered with Welsh GPs. Similarly, Richmondshire, housing a military base in Catterick Garrison, and Forest Heath in Suffolk, hosting RAF Mildenhall, exhibit lower patient-to-population ratios. Conversely, Oxford and Cambridge have higher patient-to-population ratios, attributed to students registered with local GPs but not counted as residents (the LSOA with highest residual is Oxford 006F, featuring student housing for Oxford Brookes University, as well as an International Language Campus).

Additionally, our analysis highlights broader factors contributing to healthcare inequalities. The residual values, representing deviations from the linear fit, correlate with deprivation levels ($r = .16$, $p \approx 0$ for mid-deprived areas; $r = .22$, $p \approx 0$ for highly-deprived areas), suggesting a greater burden on healthcare access in socioeconomically disadvantaged regions (e.g., one of the areas with the highest patient-to-population ratio is near Bolton, a highly deprived region in Greater Manchester). Moreover, the residual values exhibit a negative correlation ($r = -.40$, $p \approx 0$) with the percentage of White population, indicating disparities associated with ethnic backgrounds. These findings underscore the presence of healthcare access inequalities across the country, as corroborated by prior

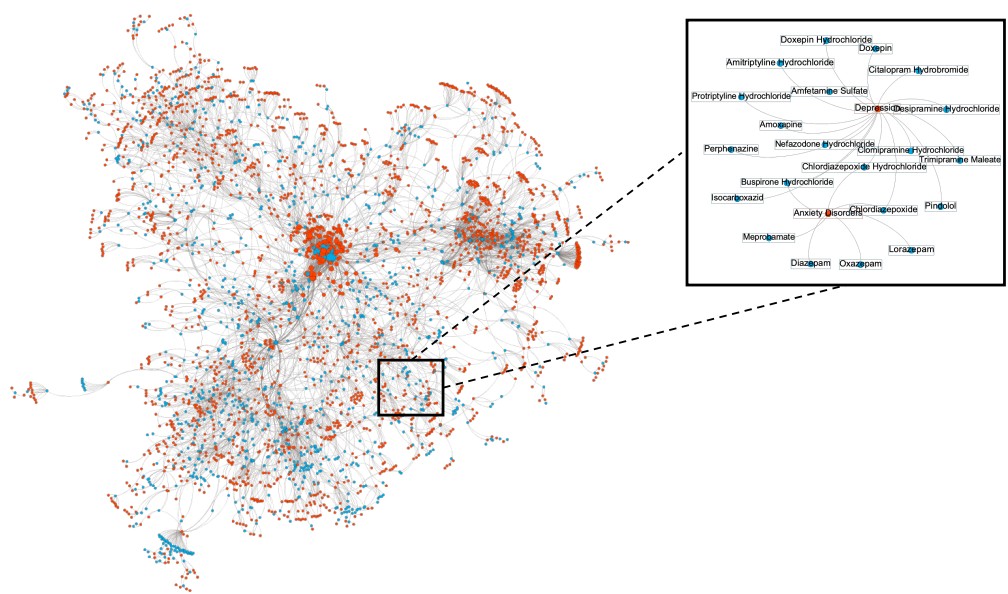

Figure 7: **Visualization of the Drugbank network:** showcasing the comprehensive interconnections between various drugs; the inset provides a zoom-in view to the subset specifically associated with *anxiety* prescriptions.

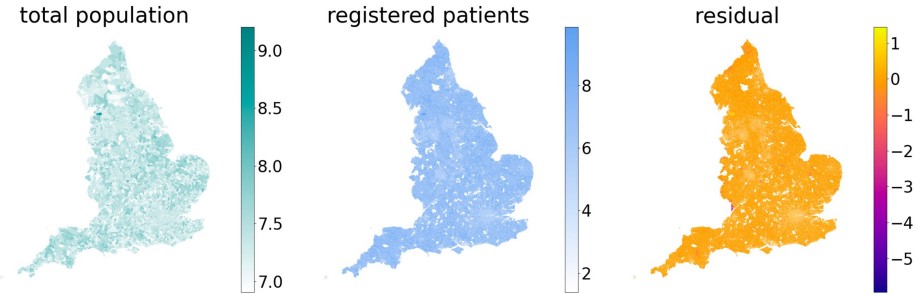

Figure 8: **Health access disparity analysis using MEDSAT.** We present a quick analysis of health access disparities by examining the relationship between the population and the number of registered patients (log-transformed), revealing a strong correlation ($r = 0.87$, $p \approx 0$). However, the residual values from a linear fit between the two highlight a clear rural-urban pattern, with urban areas showing a higher patient-to-population ratio, which correlates with the percentage of White population ($r = -0.40$, $p \approx 0$). Furthermore, the residuals show associations with mid- ($r = 0.16$, $p \approx 0$) and highly-deprived areas ($r = 0.22$, $p \approx 0$), indicating a potentially greater health-access burden in these areas. Details of the analysis are found in Appendix.

research examining various healthcare system aspects, including NHS data on waiting times, staffing, hospital activity, outcomes, and the GP Patient Survey [41], or COVID-19 hospitalisations and deaths [28].

## E.2 Predicting Prescriptions

### E.2.1 Spatial Lag Model (SLM)

The Spatial Lag Model (SLM) [3, 4], differs from the Ordinary Least Squares (OLS) model by the inclusion of the spatial lag term ($\rho W y$):

$$y = \rho W y + X\beta + \epsilon,$$

where $y$ is the outcome, $X$ is the matrix of features, and $\epsilon$ is the error term. $W$ is a matrix of the shape $N \times N$ and it contains spatial weights capturing the spatial interaction among variables, and the coefficient of spatial autocorrelation is captured by $\rho$. We fed in input to SLM the LSOA shapefiles to calculate the spatial lag term.

Table 6: **The spatial $R^2$ scores resulting from the Spatial Lag Model (SLM)**. These scores are computed across various prescription types and combinations of dataset features specifically for the year 2020.

| | metabolic | | mental | | respiratory | | |
| input | diabetes | hypertension | depression | anxiety | asthma | opioids | total |
|---|---|---|---|---|---|---|---|
| Image | 0.04 | 0.23 | 0.23 | 0.19 | 0.25 | 0.28 | 0.11 |
| Env. | 0.13 | 0.34 | 0.41 | 0.37 | 0.34 | 0.49 | 0.25 |
| Soc. | 0.33 | 0.44 | 0.50 | 0.48 | 0.38 | 0.52 | 0.31 |
| Env. + Soc. + Image | 0.38 | 0.49 | 0.54 | 0.55 | 0.43 | 0.63 | 0.38 |

The SLM results are presented in Table 6.

### E.2.2 Machine Learning Models

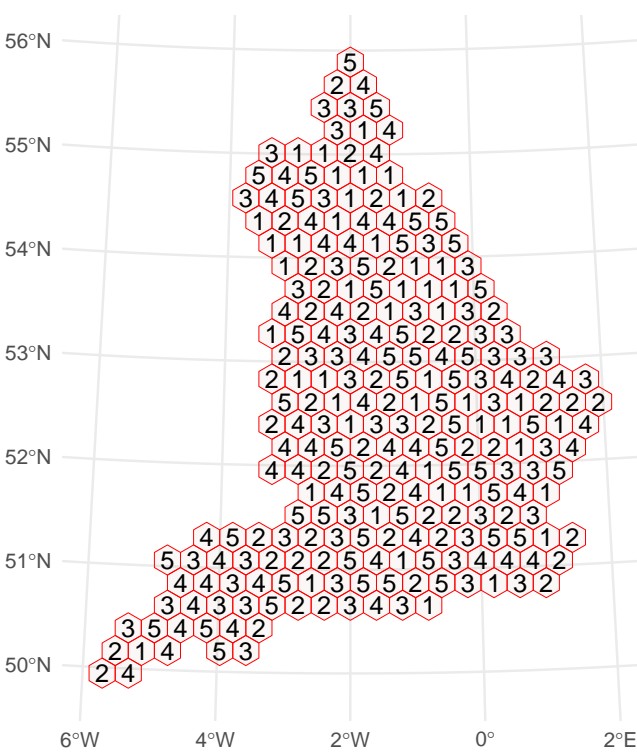

Figure 9: **Visualization of an example spatial fold created using `blockCV` spatial blocking:** we chose hexagonal blocks and randomly assigned those blocks to 5 folds consisting of an equal number of blocks. The numbers specify the fold assignment of the blocks.

**Spatial Cross-Validation**    In spatial data scenarios, traditional random cross-validation methods can inadvertently undervalue prediction errors which can lead to suboptimal model choices. Recognizing this challenge, specific validation techniques tailored for spatial modeling, such as spatial blocks

and buffers, have been introduced [23, 43]. We utilized the R package `blockCV`, which facilitates the creation of spatially or environmentally separated folds. We used `cv_spatial` function to create spatial blocks (hexagons) and relied on the interactive tool `rangeExplorer` withing `blockCV` to visualise the blocks and assess the impact of block size on the number and arrangement of blocks in the landscape. For our experiments, we chose the block size of 28km x 28km.

**Prescription Prediction**    To understand whether the prevalence scores for the different prescription types can be reliably predicted based on the environmental, sociodemographic indicators and the simple image features, we performed spatial cross-validation for the LightGBM [22] tree-based model and a Feed-Forward Neural Network (FNN). For this experiment, we dropped the instances having missing values. The hyperparameters used for LightGBM model and the FNN are provided in Tables 7 and 8, respectively. For both years in our dataset, we implemented the following evaluation procedure: We randomly created 5 splits consisting of 5 folds where each fold contains an equal number of the above-described spatial blocks. Figure 9 depicts one example of a spatial split consisting of 5 folds. For each split, we performed the standard cross-validation procedure such that 80 % of the spatial blocks are used in model training and the rest 20 % for model testing. This procedure ensures that the test set does not contain LSOAs from the geographical blocks used for model training. Further, we used half of the instances in the test fold as a validation set for early stopping to prevent overfitting, and the other half was used as a test set to evaluate the model performance.   Additionally, in each split, we performed feature standardization such that each feature has zero mean and unit variance. Figure 10 shows the average $R^2$ scores over the test sets and the corresponding standard deviation per prescription type and machine learning model. These results show that for both years, the LightGBM model consistently outperforms the FNN. Moreover, we can also see that the $R^2$ scores are similar for the different prescription types for both years except for the total prescriptions which have lower goodness-of-fit in 2019 compared to 2020.

Table 7: LightGBM hyperparameters.

| parameter | value |
| --- | --- |
| objective | regression |
| metric | rmse |
| boosting | gdbt |
| data_sample_strategy | bagging |
| num_iterations | 100 |
| learning_rate | 0.1 |
| tree_learner | serial |
| early_stopping_rounds | 10 |

Table 8: Feed-Forward Neural Network hyperparameters.

| parameter | value |
| --- | --- |
| hidden layers | 3 |
| embedding dimension | 512 |
| epochs | 100 |
| criterion | mse |
| optimizer | Adam |
| learning_rate | 0.0001 |
| weight_decay | 0.001 |
| early_stopping_rounds | 10 |

**Detailed SHAP results**    The 10 most important features as estimated by the SHAP approach [25] and the relationship between feature values and their SHAP importance for the prediction of LightGBM models for 2019, and 2020 are given in Figures 11 and 12, respectively. The LightGBM models were trained on the proposed spatial split described in Section E.2. Although these plots indicate that the majority of the relevant factors overlap for 2019 and 2020, we also see that the important factors for a condition can change in the subsequent year. For instance, the aerosols feature

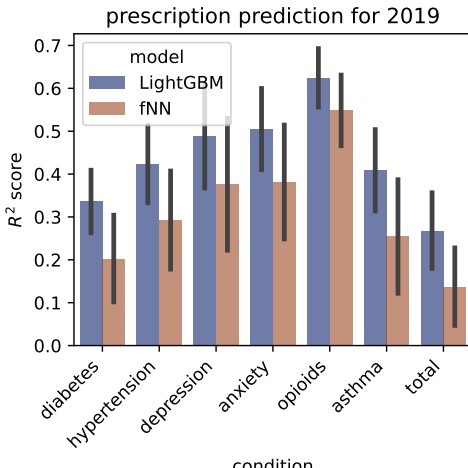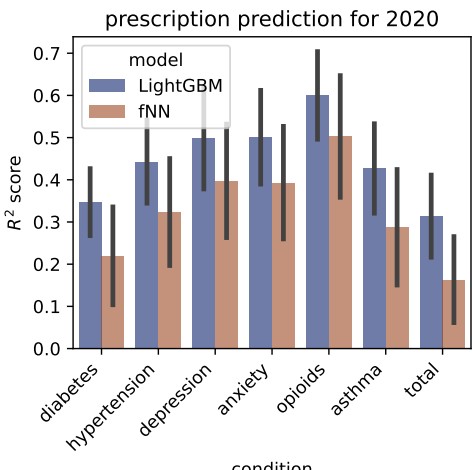

Figure 10: **Prescription prediction results on the spatial cross-validation splits of the LightGBM model and the FNN model for the different medical conditions in 2019 (left plot) and 2020 (right plot)**. The error bars indicate the standard deviation per machine learning model and medical condition and they show that the $R^2$ scores deviate strongly for the FNN model.

that does not appear among the top-15 features for predicting opioids in 2019 is the most important feature for the same condition in 2020. Also, we observe that the contribution of a feature can vary over the years. For example, while low values of thermal radiation are related with an increase in opioids prescriptions for 2019, such a relationship is not exhibited for 2020. These insights shed light on the challenges introduced with our dataset for predicting future health outcomes based on historical data.

**SHAP Depedence Plots**   The SHAP dependence plots explain the effect of a feature on the model predictions on the entire dataset while also revealing the interaction effects between the features [2]. In Figure 14, we show the SHAP dependence plots for the lightGBM models trained for predicting anxiety and opioid prescriptions. For creating the plots, we used the SHAP python library with the auto option for the interaction index that selects the feature which has the highest estimated interaction with the most important feature (according to the SHAP values). The left plot shows that LSOAs with a high number of White ethnicity population result in higher model predictions for anxiety. Further, the interaction of White ethnicity with work from home feature reveals that among the LSOAs with a large White population, lower prescriptions are predicted for those having a large percentage of people working from home. Next, the right plot shows that the model tends to predict lower prescription rates for LSOAs with high PM2.5 values while at the same time depicting no clear relationship between the PM2.5 and the wind component.

### E.3   Temporal Analyses

As Figures 15, and 16 show, both the prescription quantities for our observed outcomes, as well as the environmental point features exhibit different distributions between 2019 and 2020. People in England were prescribed more mental health-related medications, as well as *diabetes*-associated medications, and less medications associated with *asthma* and *hypertension*. Satellite data measured significantly less *NO2*, *ozone*, and *PM2.5* across England during 2020. The changes in land cover manifested by an increase in *built* environment, and a decrease in *trees* cover. Finally, *temperatures* and the amount of *solar radiation*, as well as the amounts of *wind*, as expressed through its both components, east-west and north-south, were higher in England in 2020 compared to 2019.

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
