# OpenReview forum: "MedSat: A Public Health Dataset for England Featuring Medical Prescriptions and Satellite Imagery"
_NeurIPS.cc/2023/Track/Datasets_and_Benchmarks — NeurIPS 2023 Datasets and Benchmarks Poster_

### Official Review · Reviewer_Zvuw · 2023-07-19
**MEDSAT: A Public Health Dataset for England Featuring Medical Prescriptions and Satellite Imagery**

**Rating:** 5
**Confidence:** 3
**Correctness:** The paper seems correct.

**Strengths:**

1. The paper presents a novel and comprehensive dataset that integrates medical prescriptions, environmental indicators, and sociodemographic factors for studying public and population health in England. The dataset is a valuable contribution to the research community and has many implications for machine learning and public health.
2. The paper shows the potential applications of the proposed dataset, such as predicting prescriptions, revealing health disparities, and uncovering health factors using machine learning models.

**Additional Feedback:**

Please refer to the Opportunities For Improvement.

**Clarity:**

The paper can be written more clearly. Some sections need to be rewritten e.g. section 3.2.

**Documentation:**

The paper provides sufficient detail on data collection and organization, availability and maintenance, and ethical and responsible use of the MedSat dataset. The paper states that the MedSat dataset is publicly available under a CC BY-SA 4.0 license. The code for processing the data is also available as open-source. The authors have provided a URL for accessing the data and the code.

**Ethics:**

Regarding ethical and responsible use, the paper mentions that the prescription data used in the dataset is anonymized and does not contain any personally identifiable information.

**Limitations:**

Please refer to the above paragraph.

**Opportunities For Improvement:**

1. The description of the dataset should provide more details on how the data was processed. It should contain detailed information about the environment variables.
2. There is no mention of the gender/age/ethnicity variation in the dataset. The paper should mention the distribution of these attributes.
3.  The authors should improve the baseline models. Please try to incorporate some recent baseline models.
4. The table 2 shows that combining both Env. + Soc.  features increase the $R^2$ from 0.60 to 0.64. Can we say that social features don't have an important role to play?
5. More experiments are required to conclude the importance of social features.
6. Can the author show partial dependence to capture the relationship between features and predictions?
7. Some drugs may be shared for many diseases, which may introduce noise or ambiguity in analysing the health outcomes. For example, if a drug is prescribed for asthma and COPD, it may be hard to distinguish between these two respiratory conditions based on the prescription data alone.


**Relation To Prior Work:**

The paper explicitly discusses the relation to prior work.

**Summary And Contributions:**


This paper introduces MEDSAT, a novel and comprehensive dataset that integrates medical prescriptions, environmental indicators, and sociodemographic factors for studying public and population health in England. The dataset covers all Lower Layer Super Output Areas (LSOAs) in England for the years 2019 and 2020, and includes data on prescriptions associated with seven medical conditions, 42 point features and 4 seasonal satellite images of environmental factors, and over 60 sociodemographic features. The paper also demonstrates some potential applications of the dataset, such as predicting prescriptions, revealing health disparities, and uncovering health factors using machine learning models. The paper contributes to the research community by providing a unique and valuable resource for exploring the relationships between health outcomes and environmental and sociodemographic contexts, as well as developing new techniques for modeling population health using large and complex data.

---

> ### Author Response · Authors · 2023-08-22
>
> Dear reviewer,
> we truly appreciate your insightful feedback. Below you can find the answers to your suggestions.
>
> 1. Data processing description
>    - To address the need for a description of data processing, we have enriched our discussion regarding the environmental features in the main manuscript (specifically in Section 3.2 and Figure 4) and further detailed it in the Appendix (sections D.2, D.3, and Table 3).
> Furthermore, we have made available the code we utilized for crafting environmental image composites via WASDI servers. Given that WASDI supports complimentary access for scientific projects, we believe our shared code might benefit other researchers keen on exploring this computational platform. It is worth highlighting that our endeavors in crafting a composite imagery for a single year necessitated processing over **8TB** of satellite imagery.
> Additionally, we have updated and enhanced our code documentation and user guides pertaining to the extraction of environmental point features. This can be easily accessed and employed using a complimentary Google Earth Engine account, allowing flexibility for researchers to modify it for different time frames or geographical areas.
> 2. Gender/Age/Ethnicity distributions
>     - We provided the plots for the gender/age/ethnicity distributions in the Datasheet in Appendix (in the answer to question *Does the dataset identify any subpopulations (e.g., by age, gender)?*).
> 3. Incorporation of baseline statistical models
>    - To enhance the robustness of our analyses, we have made two pivotal modifications:
> i) We have integrated a foundational model from spatial econometrics, specifically the Spatial Lag Model (SLM), as a baseline which is adept at handling data with spatial autocorrelation—characteristic of our dataset.
> ii) To ensure a more rigorous evaluation of our machine learning models, we employed spatial cross-validation.
> The results of both approaches can be seen in Section 4.1, Tables 2 and 3, respectively.
> 4. Feature importance
>     - Thank you for your keen observation. With the inclusion of spatially-aware models, specifically the SLM and ML models trained with spatial folds, the dynamics have shifted. For the SLM, sociodemographic features indeed outperform environmental ones across all prescription types. However, with the LightGBM model, this predominance is now observed only for diabetes and hypertension.
> 5. Partial dependence plots
>    - We sincerely appreciate your suggestion. In response, we have included SHAP dependence plots that are similar to partial dependence plots, but also depict the interaction effects between the features, and only consider regions of the input space in the domain of the input features [1]. This analysis can be found in Appendix E.5, specifically Figure 13.
> 6. Linkage between drugs and diseases
>    - Thank you for highlighting these areas. We are in the process of refining the Appendix to provide a detailed list of all the drug names utilized in our study. Additionally, we are addressing and clarifying the point you mentioned under the "Limitations" section. It is worth noting that our approach aligns with a substantial body of existing research that considers drug prescriptions related to specific conditions as indicators of health outcomes, which we will cite in the manuscript.
> 7. Improving text clarity
>    - Thank you for your comment. We took care to revisit sections on environmental features. We first revised section 3.2 on environmental point features, shortening, and simplifying the description. Next, we improved the description of environmental image features, including adding the diagram in Figure 3 to depict the processing we undertook.
>
> [1] https://shap.readthedocs.io/en/latest/example_notebooks/tabular_examples/tree_based_models/Census%20income%20classification%20with%20LightGBM.html

---

> > ### Author Response · Authors · 2023-08-25
> > **drugs may be shared for many diseases**
> >
> > Dear reviewer,
> >
> > We have now also addressed your point regarding the **drugs shared for many diseases**. Specifically, we:
> >
> > i) supplied the lists of drug names linked to each of the conditions highlighted in this work (please consult the Appendix section *DrugBank Network* and our GitHub repository at https://anonymous.4open.science/r/MedSat-7S/code/collate_data/NHS_prescription_parser/drug_names/); and
> >
> > ii) emphasized in the manuscript that for simplicity, we use condition names to refer to related prescriptions, i.e., we consider drugs *associated* with any of the discussed conditions, but not definitively *prescribed for* that specific condition by the GP.  This distinction is due to the absence of such specific prescription intent in the NHS prescribing dataset. Consequently, this might lead to co-prescriptions for drugs associated with multiple conditions in DrugBank. We direct you to Section 3.1 Medical Prescriptions (lines 135-146) and the Limitations section (lines 334-343) for further details.
> >
> > Thank you.

---

### Official Review · Reviewer_xh3o · 2023-07-20
**Review of MedSat, a novel and multi-modal dataset of prescriptions, environment, and sociodemographics in England**

**Rating:** 8
**Confidence:** 4
**Correctness:** The MedSat dataset is constructed in …
**Clarity:** The paper is very well written.

**Strengths:**

Please find a list of strengths below:

1. The motivations are strong for the creation of MedSat.
2. Incorporation of data prior to and during COVID allows for disentanglement of COVID effects.
3. Comprehensive discussion of related work and how MedSat complements existing resources.
4. The authors clearly describe how they curated the MedSat dataset.
5. The dataset is anonymized, alleviating potential ethical concerns.
6. Dataset use cases are clearly described.
7. The authors conduct an interesting analysis of predicting prescription rates using MedSat. In addition, the authors use SHAP to analyze the contribution of different input features. This analysis revealed interesting patterns in the dataset and serves as a fantastic example for how other researchers can use MedSat to study the interaction of health, the environment, and sociodemographic factors.
8. The authors include a compelling discussion of limitations and future work.
9. The figures are useful and visually appealing. FIgures 2, 3, and 4, for instance, do a fantastic job of demonstrating the breadth, depth, and variability of the dataset.
10. MedSat has a permissive license.
11. Code to recreate the dataset is provided.

**Additional Feedback:**

This is excellent work. I list a few minor concerns and one remark.

Minor concerns:
1. The authors write that MedSat contains prescription information for seven medical conditions. In Table 1, I see six listed: asthma, diabetes, hypertension, depression, anxiety, and opioids. Can the authors please clarify what the seventh condition is?
2. In line 156, can the authors clarify what the “previous approach” is?
3. In lines 47-48, the authors write that they present three complementary datasets. While this is true, it confused me when I first read it. I was expecting that MedSat was one dataset. The authors might consider rewording “three complementary datasets” to “a dataset with three complementary components”.

Other remarks:
1. In Table 1, it is interesting that environmental input variables consistently outperform sociodemographic input variables (in many cases by a large margin). Using the combination of environmental and sociodemographic variables leads to the best performance, but the performance is often not much greater than that of models using environmental inputs only.

**Documentation:**

The manuscript provides sufficient detail on the creation of the dataset. However, the availability and maintenance plan for the dataset is not clear to me. As a reviewer, I have access to the dataset via DropBox. I strongly suggest hosting the dataset on a platform that assigns a DOI, like HuggingFace Hub or Zenodo.

**Ethics:**

I do not suspect any ethical concerns.

**Limitations:**

Yes, the authors adequately address limitations and potential negative societal impact.

**Opportunities For Improvement:**

Please find a list of opportunities for improvement below:

1. Currently the MedSat dataset is hosted on Dropbox. I urge the authors to consider hosting MedSat on a platform that provides a DOI, like Zenodo or HuggingFace Hub. Once assigned a DOI, the dataset is more or less guaranteed to persist in perpetuity, whereas hosting on DropBox does not have any such guarantee.
2. “Opioids” is not a medical condition. Perhaps this could be renamed to pain management? Although I do admit that opioids have uses beyond pain management.
3. Can the authors please include a measure of spread (eg standard deviation) in Table 2? The values are the averages of 5-fold cross validation. I would find it useful to know the spread of the cross validation results.
4. I was not able to view the image data in MedSat. In the README of the Dropbox directory, the authors explain that the image data requires registration currently: “To address this, we are actively working on transferring the data to an external server. This will allow you to conveniently access and review the data without the need for registration. We will promptly provide you with the access link as soon as the transfer is complete”. Is there a way I as a reviewer can view the image data? I see that the image_data directory on Dropbox is empty.

**Relation To Prior Work:**

The authors clearly describe related work. It is obvious to the reader how MedSat fits into the broader field.

**Summary And Contributions:**

The authors present a rich dataset of medical, environmental, sociodemographic, and satellite image information. The authors write that MedSat is “a comprehensive resource for public and population health studies in England, encompassing medical prescription quantity per capita as outcomes and a wide array of sociodemographic and environmental variables as features.” As a reviewer, I agree with this statement. MedSat also represents a novel and useful resource for researchers. The dataset is motivated well and it appears to fill a real gap in the available resources. Limitations are also discussed appropriately. I anticipate that MedSat will enable interesting and useful analyses of the environment and health. Finally, the paper is written well and was a pleasure to read.

---

> ### Author Response · Authors · 2023-08-22
>
> Dear reviewer, thank you for your valuable feedback and patience.
>
> 1. Dataset hosting and download
>     - We have made our dataset available for download on the TUMMedia server. Along with it, we have provided easy-to-follow instructions on how to link imagery with LSOAs and process it. Please see our collective response to all reviewers for more details.
> 2. Opioids as a medical condition
>     - Thanks for noting the incorrect opioids classification. We have revised our manuscript and reclassified opioids out of the category of "mental" conditions and into a distinct group. While they are associated with the UK's opioid crisis, and they are prescribed for pain management, they do not fit within the mental category, and they are not a separate condition, as you correctly suggested.
> 3. Measure of Spread in the results
>     - We have included the measure of spread in Table 3, as requested by the reviewer.
> 4. Set of considered medical conditions
>     - We appreciate your attention to detail and highlighting this inconsistency. You are correct; the manuscript should consistently mention prescriptions for five medical conditions in addition to the total, and opioids that we now separated out, thus referencing seven prescription types. We have revised the manuscript to ensure this clarity throughout. Thank you for bringing it to our attention.
> 5. Describing the previous approach in Section 3.2
>     - We greatly appreciate your keen observation on the clarity of that sentence. We have revised it for better clarity and it now reads:
> In addition to the above-described approach for deriving preprocessed Environmental Point Features, we also directly used the different spectral bands provided by Sentinel-2, thus resulting in a more comprehensive set of Environmental Image Features.
>  6. MedSat, a dataset with four distinct components
>     - We sincerely appreciate your constructive feedback. In light of your suggestion, we have revised our description to refer to a unified MedSat dataset, which comprises four distinct components. Notably, in this revision, we have also made the environmental image features accessible. Your insights have been instrumental in refining our presentation, and we thank you for that.
> 7. Relevance of the sociodemographic and the environmental features
>     - We genuinely appreciate your keen observation. With the inclusion of spatially-aware models, specifically the SLM and ML models trained with spatial folds, the dynamics have shifted. For the SLM, sociodemographic features indeed outperform environmental ones across all prescription types. However, with the LightGBM model, this predominance is now observed only for diabetes and hypertension.

---

> > ### Comment · Reviewer_xh3o · 2023-08-29
> >
> > Dear Authors,
> >
> > Thank you for your responses to my review. You have addressed all of my comments. I will maintain my score of 8. Great work!

---

### Official Review · Reviewer_nQhp · 2023-07-21
**MedSat addresses an important gap in the literature, however, the documentation, reproducibility, accessibility, and analysis need improvement.**

**Rating:** 7
**Confidence:** 5

**Strengths:**

- The dataset facilitates accessing prescription data from the NHS in the UK, which needs substantial processing, making it more useful for future research.
- Imagery data is hard to process and, therefore, seldom used. Making it more accessible can positively impact public health studies.

**Additional Feedback:**

Question:
- Can causal conclusions be derived from this dataset? Are there potential missing confounders?
- Should conclusions from this specific dataset inform policy in the UK? Can they be transferred elsewhere?
- Why just 2019, 2020, and not earlier/later years?
- What other diseases could be considered? Air pollution has been strongly linked to cardiovascular diseases [1].


[1] Peng, R.D., Chang, H.H., Bell, M.L., McDermott, A., Zeger, S.L., Samet, J.M. and Dominici, F., 2008. Coarse particulate matter air pollution and hospital admissions for cardiovascular and respiratory diseases among Medicare patients. Jama, 299(18), pp.2172-2179.

**Clarity:**

The paper is mostly very clear.
The temporal domain (season per year) should be mentioned more clearly and justified earlier in the text.

**Correctness:**

I am concerned about the correctness of the statistical analysis. See my note on the Opportunities for improvement regarding the use of cross-validation.

**Documentation:**

- The bar for reproducibility should be high in this Datasets track. I could not run the notebooks provided in the Dropbox folder (issues with dependencies, setting up the API, and it was unclear to me where/how to download the auxiliary data).
- Conda environments or containers (hopefully both) are recommended. Any steps requiring human interaction (such as signing up for APIs and manually downloading data) should be avoided when possible or thoroughly documented.
- I could not access the imagery data. At least a sample should be provided with examples of how to access it and manipulate it.
- The main documentation missing is that of how a new user can access and download your data, which should follow the FAIR principles, including machine-leaning readiness. The authors already mention in the appendix that they will share the data with a service like figshare. This is a good idea. I recommend sharing a sample using and writing an example documentation.

**Ethics:**

I do not suspect ethical concerns.

**Limitations:**

The discussion is reasonable. See also questions in the additional feedback section. The literature on environmental exposures (including air quality) in hospitalization is massive, and it makes sense to focus on those related to satellite imagery.

**Opportunities For Improvement:**

- The statistical analysis must exemplify the advantage of combining satellite imagery with standard public/environmental health point datasets. My understanding is that the GBMs/NNs analysis in the current version completely ignores the imagery data. The only way in which  I see the satellite imagery being sued is to draw the maps in Figure 3. Unfortunately, I don't see this use case as much more useful than simply using Google Maps to produce a similarly-looking map.
- High R2 scores with spatial data are very often caused by spatial autocorrelation when using random train/test splitting (essentially, there's data in the validation set that is almost identical to the training set). The authors should implement a form of block-buffered cross-validation (e.g., [1]). Further, due to the nature of this track, they should make a recommendation on how to implement it in the future. I saw some notes about geographic splitting in the appendix but could not find them in the code or main text.
- The reproducibility and accessibility of the data should be improved. While the authors can't share the full data at the moment, they should at least provide a sample of the imagery data and exemplify how to access it. I find the current Dropbox strategy to be insufficient (although the authors do promise to use a service like Figshare). The current Dropbox does not contain any imagery data and does not contain sufficient documentation (see my other notes in the Documentation section).

[1] Le Rest, K., Pinaud, D., Monestiez, P., Chadoeuf, J. and Bretagnolle, V. (2014), Spatial leave-one-out cross-validation. Global Ecology and Biogeography, 23: 811-820. https://doi.org/10.1111/geb.12161

**Relation To Prior Work:**

To my knowledge, prior work is discussed appropriately for this track. Table 1 is very useful.

**Summary And Contributions:**

The MedSat study combines demographic data with satellite imagery to analyze public health trends in 2019 and 2020. While the premise is interesting, the paper currently lacks important elements such as reproducibility and proper documentation and accessibility. Additionally, the analysis does not use appropriate cross-validation methods for spatial data, thereby invalidating the author's recommended use case and evaluation metric. Their analysis does not provide a strong justification for using imagery data and environmental health variables together, which is the core premise of the paper.  As a result, I cannot recommend accepting the paper in its current state.

---

> ### Author Response · Authors · 2023-08-22
>
> Dear reviewer, we are very grateful for your insightful comments.
>
> 1. Benefits of using satellite imagery
>     - To incorporate the satellite imagery into our modeling pipeline, we have introduced a script that extracts LSOA-centric images from the seasonal Sentinel-2 composite tiles and outputs descriptive statistics of the Sentinel-2 bands per LSOA. The predictive accuracy of the models trained on these low-level features has been showcased in Tables 2 and 3 (albeit lower compared to that of the high-level sociodemographic and environmental features). Further, in Section *Describing Health of Environment Using Visual Concepts* we show that the model trained on simple image features associates aerosol air pollution with higher predicted prescription values. Even though we applied a baseline approach, our findings indicate that the descriptive image features can enable the machine learning model to associate plausible links between environmental conditions and population health.
> 2. Statistical analysis and spatial autocorrelation
>     - To address spatial autocorrelation, we implemented the traditional spatial econometrics model Spatial Lag Model (SLM) and the results have been included in Table 2 of the main manuscript. Next, based on your recommendation, we also adjusted the LightGBM model for our spatial context. In the main manuscript, specifically in the third paragraph of the section titled *Predicting Prescriptions* and further in the Appendix section E.1.1 *Spatial Cross-Validation,* we highlight our use of the `blockCV` R package to create spatial folds for our dataset. Our updated results using LightGBM with spatial-folds cross-validation can be found in Table 3 of the main manuscript. Furthermore, we've included these spatial folds in our repository, offering a proposed method for data partitioning in subsequent research.
> 3. Data Access and Reproducibility
>     - Please see our common answer to all the reviewers detailing how our data can now be accessed on the Technical University Munich data-sharing server, as well as how we have improved the documentation.
> 4. Air quality and health
>     - As our SHAP results from the models using the environmental point features show, we do find associations between air quality and the MedSat outcomes. For example, PM2.5 is found associated with higher diabetes and hypertension prescriptions, and higher ozone concentration is linked with higher anxiety prescriptions. Moreover, our results with the newly introduced image features reveal that the Sentinel-2 band B01, which is sensitive to the presence of aerosols, is linked with higher prescriptions of opioids. We believe that MedSat can help to unveil further valuable links between air quality and health outcomes.
> 5. Temporal domain
>     - To address your suggestions, we have amended the conclusion of the Introduction to explicitly mention that our dataset comprises four complementary segments spanning the years 2019 and 2020. Additionally, we have updated Figure 1 to clearly indicate that *MedSat* provides two annual snapshots, specifically for 2019 and 2020, across all its four data segments.
> 6. Code reproducibility
>     - We wholeheartedly agree, and took great care to make our work reproducible: we restructured the repository, improved the README instructions, provided 'conda' environment .yml files for different modules in our repository, included clear instructions for dataset download, and a guide, including snapshots, for Google Earth Engine account creation (which is needed only for running the extraction of environmental point features).
> 7. Causality and UK Health Policy insights
>     -  We are conducting a study whose primary objective is to delve into identifying causal relationships. Our strategy involves feature selection to incorporate critical covariates for specific conditions while minimizing correlated features. Additionally, we are exploring cutting-edge causal modeling techniques, such as causal graph neural networks. We firmly believe in the potential of MedSat. While it might not directly influence policy, its findings could serve as valuable precursors. Should MedSat reveal novel associations or causal links between prescriptions and the environment, these insights could lay the groundwork for specialized cohort studies. By further elucidating these relationships, we envision that this information could subsequently inform and shape UK policy. Gathering data from earlier years presented us with two significant challenges. First, obtaining annual outcomes needs the integration of prescription data with annual patient counts from a separate NHS source matched to a different LSOA file version. The manual nature of this task made it challenging for us to expand the outcome duration. Second, our decision to emphasize advanced and high-resolution products meant that many of the satellite products, from which we extracted environmental indices, were not available beyond the recent 2-3 years.

---

> > ### Comment · Reviewer_nQhp · 2023-08-28
> >
> > The revisions addressed my concerns with cross-validation. I will raise my score accordingly.

---

### Official Review · Reviewer_AAen · 2023-07-23
**Potentially nice dataset paper, but some gaps in the analyses and data availability**

**Rating:** 7
**Confidence:** 4

**Strengths:**

- Carefully curated datasets for social determinants of health are critical valuable for health research.
- Satellite imaging is an exciting modality to link in a geospatial/temporal setting

**Additional Feedback:**

N/A

**Clarity:**

- Yes the paper is written clearly, modulo lack of clarity on how imaging is accessible and linked to the release dataset.

**Correctness:**

- The paper appears correct.

**Documentation:**

- The authors provide their code, but there is lack of clarity of the imaging data side of the release (is it available for download, is WASDI hosting it, does the dataset change in anyway over time, etc.) Other considerations: sites like FigShare have max limits at 20GB and require using different services (FigShare+) to support very large files.

**Ethics:**

- No ethics reviews.

**Limitations:**

- The linkage of drugs to condition is coarse and imperfect, e.g., why are opioids binned under the "mental" class?  The appendix only seems to include drugs associated with diabetes so it's not possible to sanity check some of the assignments. The appendix acknowledges "a single API or medical drug can be used to treat various conditions" That seems a serious limitation of the paper's analyses for any conclusion involving a specific disorder.

**Opportunities For Improvement:**

- Are Sentinel-2 images available as part of this dataset or just linkages to images? Reading the code in the repo, it wasn't obvious whether users of this dataset could download imaging as part of this dataset vs. use data on WASDI. The appendix suggests all imaging is hosted by WASDI platform. Will the imaging data be available to reviewers?
- The dataset is considerably less useful IMO if there isn't a mechanism to download data to their research compute environment. If the data can be downloaded, the code should include a clear script or set of tools for doing so. If the data cannot be downloaded, the manuscript should be clear the dataset is for linking to imaging available via API, not a dataset of satellite images itself.
- A core benefit of this paper is the linkage to imaging data, yet pixel analyses are not really explore, beyond some summary factors around climate scores, This is a significant limitation. Lines 235-243 make mention of some potential benefits of visual concepts, but there are no experiments exploring this.
- Given the temporal nature of the dataset (pre/during COVID), more experiments specifically looking at this change point would improve the manuscript.
- Baseline analyses are quite simple (correlational analyses using GBM)
- Given the well known correlations between geographic location / social determinants of health and future outcomes, R^2 scores are a best a sanity check on some of the assumptions of the allocation of people to LSOAs.
- Some of the phrasings of correlation findings, e.g., line 227 "further, although high ozone values contribute to increased prescriptions" I feel greatly overstate the strength of the analysis.
- The supplement indicates 10% of the LSOA instances have missing values, yet Figure 2 suggest complete coverage. I would highlight missing values if possible using black.

**Relation To Prior Work:**

- The authors do a nice job enumerating existing datasets and prior work here.

**Summary And Contributions:**

This manscript presents MEDSAT, a dataset of public health variables linked to geospatial and sociodemographic features for LSOA geographic units in the United Kingdom. The authors link high-resolution satellite imagery, sociodemographic variables, and medical prescriptions at the level to LSOA regions to facilitate better analyses of the role of social determinants of health on outcomes. Several baseline correlational analyses with LSOAs are conducted after aggregating and cleaning up the various extracted variables. The authors state their contributions as a public dataset of satellite imaging, health indicators at high geographic resolution where indicators are collected directly via the NHS vs standard surveys. The manuscript explores several use cases (revealing health disparities, uncovering health factors using SHAP values) to motivate the benefit of their dataset.

---

> ### Author Response · Authors · 2023-08-22
>
> Dear Reviewer, we express our gratitude for your insightful feedback.
>
> We respond to the provided suggestions individually below.
>
> 1. Acess to Sentinel-2 images and dataset download
>      - Regarding the accessibility of our dataset, the vast amounts of the data that we processed posed challenges in rendering early access for reviewers. Specifically, the Sentinel-2 image data was processed on WASDI servers. After the processing was finalized, we transferred the Sentinel-2 images and the other point features to TUMMedia, which facilitates the open-source publication of research data. We direct your attention to our consolidated response to all reviewers, where we have provided a link accompanied by requisite credentials to access the satellite imagery and the other features in our dataset.  Along with it, we have provided easy-to-follow instructions on how to link imagery with LSOAs and process it.
> 1. Documentation for reproducing the dataset
>     - We appreciate this feedback and addressed it in our summary response.
> 2. Linkage and benefits of introducing image data
>    - We have introduced a script that extracts LSOA-centric images from the seasonal Sentinel-2 composite tiles and outputs descriptive statistics of the Sentinel-2 bands per LSOA. The potential of using this data for modeling population health has been showcased in our revised manuscript. This augmented analysis can be found in Tables 2 and 3 and in the section titled *Describing Health of Environment Using Visual Concepts*. The analysis in this section illustrates that the aerosol band which quantifies air pollution can be associated with higher predicted prescription values. Even though we used a baseline approach for extracting image features, these findings indicate that the descriptive image features can enable the machine learning model to associate plausible links between environmental conditions, visual satellite image features, and population health.
> 3. Baseline analysis and geographical correlations
>     - Thank you for your insightful comments. In response, we have strengthened our analyses in the following two ways:
> i) As part of our enhanced methodology, we integrated results from the Spatial Lag Model (SLM). Recognized for its proficiency in spatial econometrics, the SLM adequately adjusts for the influences of spatial autocorrelation in predictions.
> ii) We also re-evaluated our validation approach for the ML Models (LightGBM and a feed-forward neural network). Replacing our previous random cross-validation, we have now implemented a cross-validation methodology specifically tailored for spatial data, namely, spatial blocking. Our reassessment indicates consistent results between the SLM baseline model and our spatially cross-validated LightGBM model. These coherent results from both approaches further confirm the relevance of the sociodemographic and environmental features in our dataset for modeling population health.
> 4. Phrasing of correlation findings
>     - We appreciate your remark regarding our oversight in characterizing SHAP plot outcomes. The terms 'effects' were inappropriately utilized in lieu of 'associations'. We addressed this remark in the section titled *Uncovering Health Factors*.
> 5. Missing values:
>    - We have now updated Figure 2 to show the missing values in black, as suggested by the reviewer. Depending on the specific analyses intended, the missing value rate across the whole of England stands at 5.7% LSOAs (constrained by the missing values for the outcomes) up to 13% if all of the components of our dataset intend to be used (due to missing environmental indicators from two sources, that are ECMWF and CAMS). Notably, we possess sociodemographic and image features data for all LSOAs. Please refer to Appendix for the details.
> 6. Linkage of drugs to conditions
>     - Thank you for highlighting these areas. We are in the process of refining the Appendix to provide a detailed list of all the drug names utilized in our study. Additionally, we are addressing and clarifying the point you mentioned under the "Limitations" section. We will finalize this part of the response within two days. It is worth noting that our approach aligns with a substantial body of existing research that considers drug prescriptions related to specific conditions as indicators of health outcomes, that we will cite in the manuscript. Thanks for noting the incorrect opioid classification. We have reclassified opioids out of the "mental" category and into a distinct group. While they are associated with the UK's opioid crisis, they are prescribed for pain management, and do not fit only within the mental category.

---

> > ### Author Response · Authors · 2023-08-25
> > **Linkage of drugs to conditions**
> >
> > Dear reviewer,
> >
> > We have now also addressed your point regarding the **Linkage of drugs to conditions**. Specifically, we:
> >
> > i) supplied the lists of drug names linked to each of the conditions highlighted in this work (please consult the Appendix section *DrugBank Network* and our GitHub repository at https://anonymous.4open.science/r/MedSat-7S/code/collate_data/NHS_prescription_parser/drug_names/); and
> >
> > ii) emphasized in the manuscript that we consider drugs *associated* with any of the discussed conditions, but not definitively *prescribed for* that specific condition by the GP. This distinction is due to the absence of such specific prescription intent in the NHS prescribing dataset. Consequently, this might lead to co-prescriptions for drugs associated with multiple conditions in DrugBank. We direct you to Section 3.1 Medical Prescriptions (lines 135-146) and the Limitations section (lines 334-343) for further details.
> >
> > Thank you.

---

> > ### Author Response · Authors · 2023-08-28
> > **Temporal Analyses: temporal nature of the dataset (pre/during COVID)**
> >
> > Dear reviewer,
> >
> > To your comment about the temporal nature of the dataset (pre/during COVID), we have now added analyses of the differences in distributions between the two years for MedSat outcome and environmental features. Please refer to Section 4.3 in the main manuscript and Appendix Section E.6, as well as Figures 14 and 15. These analyses reveal how COVID has had different effects on various prescription types (a decrease in prescriptions for asthma and hypertension and an increase in mental conditions and diabetes). Similarly, the environmental changes are evident; notably, MedSat shows a significant decrease in air pollutants in 2020 as measured through NO2, PM2.5, and ozone.
> >
> > Kind regards

---

> > > ### Comment · Reviewer_AAen · 2023-08-28
> > >
> > > Thank you to the authors for their additions to manuscript and addressing my concerns. I have raised my score. Some additional minor comments.
> > > - Figure 5: Hard to parse (have to refer back to Fig 1 for band id definitions and color meanings). It would be nicer to have a legend in the figure itself.
> > > - The updated manuscript is now 11 pages long (vs. the 9 page limit). The authors should move some details to the appendix or otherwise trim text.

---

> > > > ### Author Response · Authors · 2023-08-29
> > > >
> > > > Dear reviewer,
> > > >
> > > > Thank you for your feedback. We updated the caption of Figure 5 by stating the meaning of the bands and the colors they represent in the image. Regarding the length of the manuscript,  we are working on shortening the text and will provide a shortened version once all comments from the reviewers are finalized.

---

### Author Response · Authors · 2023-08-22
**summary of revision**

We thank to all the reviewers for their invaluable and helpful comments. We have thoroughly revised the paper and the repository according to the reviewers' suggestions.

First, we have enabled access to all of our data, including environmental image features on TUMMedia, a data sharing service from the Technical University of Munich. The login details are below:

https://mediatum.ub.tum.de/1714817

*Login*: obadic-1714817-review

*Pass*: PuE1&L!maQ;81Hfd-DwPsNa19

The dataset is released under the CC BY-SA 4.0 license.


Second, to enhance the robustness of our analyses, we have made two pivotal modifications:
i) We have integrated a foundational model from spatial econometrics, specifically the Spatial Lag Model (SLM), which is adept at handling data with spatial autocorrelation—characteristic of our dataset.
ii) To ensure a more rigorous evaluation of our machine learning models, we employed spatial cross-validation.

Third, we have meticulously updated and refined the documentation associated with our repository, encompassing both data retrieval and processing. For every module that contributes to the creation of our dataset's four distinct components, we provide a Conda environment specification, which allows for the reproducibility of our code. Moreover, we have mirrored the dataset's structure on TUMMedia within our repository. This design provides users with a clear understanding of the appropriate location for each downloaded data file, thus enabling them to easily execute our code and reproduce the results.

Fourth, to underscore the significance of our environmental image features, we have extracted descriptive statistics from each spectral band of the LSOA image pixels. This procedure is described in Section 3.2 and visualized in Figure 3 in the main manuscript. Next, we used these extracted descriptive statistics from the image bands as input features for the SLM and the machine learning models.
While utilizing these basic, low-level features reduces the model's predictive capacity compared to when high-level sociodemographic and environmental features are used, we found that the image features still facilitate the model in discerning plausible relationships between environment and prescriptions. Such findings support the argument that when image data is combined with advanced methodologies like self-supervised learning or other deep learning feature extraction techniques, there is substantial potential to gain deeper insights into health outcomes.

---

### Decision · Program_Chairs · 2023-09-22

**Decision:**

Accept (Poster)

**Comment:**

The paper provides a well-written blueprint to repurpose satellite images for public health use by linking them with other data sources.

The reviewers described a number of strengths of this work. For example, the dataset is well-motivated and valuable for health research, the combination of satellite imaging and additional data sources is well done, and the substantial processing of the data will be beneficial for future research.

The authors made significant effort to address all reviewer concerns during the discussion period. Zvuw did not give a positive rating, but this does not take into account the author discussion and is thus weighted less.

Overall, I recommend acceptance.